# First- and Second-Order Bounds for Adversarial Linear Contextual Bandits

Julia Olkhovskaya[1]    Jack Mayo[2]    Tim van Erven[2]    Gergely Neu[3]    Chen-Yu Wei[4]

[1]Department of Intelligent Systems, Delft University of Technology, Delft, The Netherlands[*]
[2]Korteweg-de Vries Institute for Mathematics, University of Amsterdam, Amsterdam, The Netherlands
[3]AI group, DTIC, Universitat Pompeu Fabra, Barcelona, Spain
[4]MIT Institute for Data, Systems, and Society, Massachusetts Institute of Technology, Cambridge, MA, USA

## Abstract

We consider the adversarial linear contextual bandit setting, which allows for the loss functions associated with each of $K$ arms to change over time without restriction. Assuming the $d$-dimensional contexts are drawn from a fixed known distribution, the worst-case expected regret over the course of $T$ rounds is known to scale as $\tilde{O}(\sqrt{KdT})$. Under the additional assumption that the density of the contexts is log-concave, we obtain a second-order bound of order $\tilde{O}(K\sqrt{dV_T})$ in terms of the cumulative second moment of the learner's losses $V_T$, and a closely related first-order bound of order $\tilde{O}(K\sqrt{dL_T^*})$ in terms of the cumulative loss of the best policy $L_T^*$. Since $V_T$ or $L_T^*$ may be significantly smaller than $T$, these improve over the worst-case regret whenever the environment is relatively benign. Our results are obtained using a truncated version of the continuous exponential weights algorithm over the probability simplex, which we analyse by exploiting a novel connection to the linear bandit setting without contexts.

## 1 Introduction

The contextual bandit problem is a generalization of the multi-armed bandit setting in which a learner observes relevant contextual information before choosing an arm. The goal of the learner is to minimize the excess cumulative loss of the chosen arms compared to the best fixed policy for mapping contexts to arms. This framework addresses a broad range of important real-world problems like sequential treatment allocation (Tewari and Murphy, 2017), online recommendation (Beygelzimer et al., 2011) or online advertising (Li et al., 2010), and is actively used in practice (Agarwal et al., 2016). Numerous variants of the setting have been studied, which differ in the assumptions they make about the losses and the contexts. In this paper, we focus on the recently introduced setting of Neu and Olkhovskaya (2020) where the contexts are finite-dimensional i.i.d. random vectors, and the losses are time-varying linear functions of the context that may potentially be generated by an adversary. In this setting, the worst-case rate for the expected regret is known to be $\tilde{O}(\sqrt{T})$ for time horizon $T$ (Neu and Olkhovskaya, 2020).

Our main contribution is to replace the worst-case rate by adaptive bounds. Specifically, we obtain a bound of $\tilde{O}(\sqrt{V_T})$ in terms of a quadratic measure of variance $V_T$ for the losses of the algorithm, and a bound of $\tilde{O}(\sqrt{L_T^*})$, where $L_T^*$ is the cumulative loss incurred by the optimal policy. Such bounds in terms of $L_T^*$ or $V_T$ are generally referred to as *first-order* and *second-order bounds*, respectively,

---

[*]Work was done when the author was affiliated with Vrije Universiteit Amsterdam.

37th Conference on Neural Information Processing Systems (NeurIPS 2023).

and have been extensively studied in the bandit literature. They can lead to much stronger guarantees in the often realistic case when $T$ is large, but the losses vary little or when there exists a policy with very low cumulative loss.

Worst-case guarantees in terms of $T$ have first been proved for the contextual bandit problem with finite policy classes by Auer et al. (2002b), with further improvements by Beygelzimer et al. (2011). These methods can deal with adversarial losses and contexts, but only work for finite policy classes and have run-time scaling linearly with the size of the class—which is generally unacceptable in practice. This latter challenge has been addressed by a line of work culminating in Agarwal et al. (2014), which only requires access to an optimization oracle over the policy class. Their results, however, remain restricted to i.i.d. contexts and losses. An alternative line of work has been initiated by Auer (2002); Chu et al. (2011); Abbasi-Yadkori et al. (2011), who studied the special case of i.i.d. *linear* loss functions with changing decision sets. The case of i.i.d. contexts and adversarial linear losses has first been studied by Neu and Olkhovskaya (2020).

Improvements of worst-case guarantees of order $\sqrt{T}$ to first-order bounds scaling with $\sqrt{L_T^*}$ have been known for a variety of bandit settings since the works of Stoltz (2005); Allenberg et al. (2006), and Neu (2015). Regarding contextual bandits, the COLT 2017 open problem of Agarwal et al. (2017) asks for efficient algorithms that achieve first-order bounds for large, but finite, policy classes, either when both contexts and losses are i.i.d. or when both are fully adversarial. First to answer the open problem were Allen-Zhu et al. (2018), who obtained an optimal first-order regret guarantee for adversarial losses and contexts, but with an algorithm that is inefficient for large policy classes. Foster and Krishnamurthy (2021) provide the first efficient algorithm for the non-adversarial setting where the loss function is fixed over time and one has access to an oracle that can solve various optimization tasks over the policy class. We improve on these works in terms of the computational efficiency of our algorithm and by allowing the loss function to vary adversarially over time, although we do rely on the extra assumption that the loss functions are linear.

Another relevant framework is the adversarial linear bandit setting (without contexts), where there also exist adaptive results (Bubeck et al., 2019; Lee et al., 2020; Ito et al., 2020). While conceptually related, an important distinction is that the linear bandit setting assumes a fixed decision set, whereas reducing the linear contextual bandit problem to a linear bandit problem requires the use of decision sets that change as a function of the contexts.

**Main Contributions.** We consider a $K$-armed linear contextual bandit problem with $d$-dimensional contexts over $T$ rounds. The contexts are assumed to be drawn i.i.d., but the linear loss functions mapping contexts to losses for the arms are chosen by an adaptive adversary. The aim of the learner is to minimize their regret, which is the gap between the expected cumulative loss of the learner and the expected cumulative loss of the best fixed policy $\pi_T^*$ chosen in full knowledge of the sequence of losses. In this setting, $\pi_T^*$ is known to be a linear classifier, i.e. it chooses the arm with smallest predicted loss, where the predictions are fixed linear functions of the context (see Section 2). The goal is therefore to compete with all linear classifiers. We first obtain the following second-order bound on the expected regret

$$R_T = \tilde{O}\Big(K\sqrt{dV_T}\Big), \tag{1}$$

where $V_T$ is defined in (5) as a measure of the cumulative second moments of the losses for the arms played by the algorithm. Following Ito et al. (2020), we allow these moments to be centered around optimistic estimates that can further improve the bound when available or can simply be set to zero when they are not. We further obtain a first order bound of the form

$$R_T(\pi_T^*) = \tilde{O}\Big(K\sqrt{dL_T^*}\Big). \tag{2}$$

The second-order bound is obtained using a truncated version of the continuous exponential weights algorithm over the probability simplex, similar to the algorithm for linear non-contextual bandits of Ito et al. (2020), and the first-order bound may be obtained as a corollary. As discussed in Section 3.3, the computational complexity of this method is dominated by two steps that together require $\tilde{O}(K^5) + (d/\epsilon)^{O(1)}$ per round for approximation up to precision $\epsilon > 0$, which is computationally feasible for moderate $K$ and $\epsilon$. Both results are not strict improvements on the worst-case rate of $\tilde{O}(\sqrt{KdT})$ by Neu and Olkhovskaya (2020): first, they have a slightly worse dependence on $K$. We consider this a price worth paying for the first adaptive bounds in this setting. Second, they require the extra

assumption that the distribution of the contexts is *log-concave*. Although log-concavity is weaker than assuming the contexts follow e.g. (truncated) Gaussian distributions, we conjecture that it may not be necessary to obtain a computationally efficient algorithm. This conjecture is based on the observation that there exists in fact an easy way to obtain at least the first-order bound (2) without the log-concavity assumption, but with an algorithm that has no hope of being efficiently implemented. As described in Section 2.2, this is possible by running the MYGA algorithm (Allen-Zhu et al., 2018) on $O(\frac{T}{dK^2})^{Kd}$ experts that cover the set of linear classifiers to sufficient precision. The run-time of this approach is prohibitive, because it scales linearly with the number of experts, which is a large polynomial in $T$.

**Techniques.** The LinExp3 method of Neu and Olkhovskaya (2020) is based on an adaptation of the classic Exp3 algorithm for regular multi-armed bandits (Auer et al., 2002a). A natural approach would therefore be to replace the Exp3 component in LinExp3 by a method with first-order guarantees for the multi-armed bandit setting, but, as discussed in Section D, this leads to difficulties controlling the variance. Instead of building on Exp3, we therefore follow the perhaps surprising approach of building our algorithm on *continuous exponential weights* over the probability simplex (van der Hoeven et al., 2018). In particular, our approach is based on a combination of the recently proposed techniques of Ito et al. (2020) for linear bandits with tools designed by Neu and Olkhovskaya (2020) to deal with the contextual case.

**Outline.** The rest of the paper is organized as follows. After describing the setting in the next section, we state a formal version of the simple first-order bound that can be obtained using the MYGA algorithm (Theorem 2.1). This is followed by Section 3, which states our main results corresponding to the regret bounds in Equations 1 and 2. Section 4 then gives a high-level overview of the proofs, with pointers provided to the details in the appendix. Finally, Section 5 concludes with discussion.

## 2 Preliminaries

**Notation** Let $\Delta^K = \{w \in \mathbb{R}^K | w_1 \geq 0, \ldots, w_K \geq 0, \sum_{a=1}^K w_a = 1\}$ denote the $(K-1)$-dimensional probability simplex. For any positive semi-definite matrix $M \in \mathbb{R}^{d \times d}$, $\|v\|_M = \sqrt{v^\intercal M v}$ denotes the corresponding Mahalanobis norm, and for any positive integer $n$, we abbreviate $[n] = \{1, \ldots, n\}$.

### 2.1 Setting

We consider the setting of (Neu and Olkhovskaya, 2020), in which there is an interaction between a learner and an unknown environment. This interaction proceeds in rounds indexed by $t \in [T]$, such that for each $t$:

1. The environment commits to $[K]$ parameter vectors $\theta_{t,1}, \ldots, \theta_{t,K} \in \mathbb{R}^d$ without revealing any to the learner.

2. A context vector $X_t \in \mathbb{R}^d$ is drawn i.i.d. from some fixed distribution $\mathcal{D}$ according to $X_t \sim \mathcal{D}$, and revealed to the learner.

3. The learner commits to an action $A_t \in [K]$, and incurs the loss $\ell_t(X_t, A_t)$, where $\ell_t(X, a) = \langle X, \theta_{t,a} \rangle$.

The environment is allowed to randomize its choices of $\theta_{t,a}$. These must be independent from the context $X_t$ in round $t$, but they may depend on previous contexts $X_s$ and actions $A_s$ for $s < t$.

We write $\pi_t(a|X_t)$ for the policy of the learner in round $t$ conditional on observing context $X_t$, so that $A_t \sim \pi_t(X_t)$, and we use the following notation for the expected cumulative losses of the algorithm and policy $\pi$, respectively:

$$L_T = \mathbb{E}\left[\sum_{t=1}^T \ell_t(X_t, A_t)\right], L_T^\pi = \mathbb{E}\left[\sum_{t=1}^T \ell_t(X_t, \pi(X_t))\right].$$

Let $\Pi$ be the set of all all stationary deterministic policies $\pi : \mathbb{R}^d \to [K]$, we define the optimal policy $\pi^*$ as $\pi^* = \arg\min_{\pi \in \Pi} L_T^\pi$. Then the learner's goal is to compete with policy $\pi^*$, as measured by the expected regret:

$$R_T = L_T - L_T^{\pi^*} = \mathbb{E}\left[\sum_{t=1}^T \left\langle X_t, \theta_{t,A_t} - \theta_{t,\pi^*(X_t)} \right\rangle\right],$$

where the expectation is taken over each $X_t \sim \mathcal{D}$, and any randomness applied by the learner or environment in their respective choices. Using the linearity of the loss functions it can be shown that the optimal policy is always a linear classifier (Neu and Olkhovskaya, 2020):

$$\pi_T^*(x) = \arg\min_a \left\langle x, \sum_{t=1}^T \mathbb{E}[\theta_{t,a}] \right\rangle.$$

We may therefore restrict attention to competing with policies of the form

$$\pi_\beta(x) = \arg\min_a \langle x, \beta_a \rangle \qquad (\beta \in \mathbb{R}^{K \times d}). \tag{3}$$

For deriving our technical results, it will be useful to define the filtration $\mathcal{F}_t = \sigma(\{X_s, A_s : s \le t\})$, and the notations $\mathbb{E}_t[\cdot] = \mathbb{E}[\cdot \mid \mathcal{F}_{t-1}]$ and $\mathbb{P}_t[\cdot] = \mathbb{P}[\cdot \mid \mathcal{F}_{t-1}]$.

**Assumptions** Following Neu and Olkhovskaya (2020), we assume that $\|X_t\| \le \sigma$, $\|\theta_{t,a}\| \le R$ and $\ell_t(x, a) \in [-1, 1]$ almost surely. In addition, the covariance matrix $\Sigma = \mathbb{E}[XX^\top]$ of the context distribution is assumed to be positive definite, with smallest eigenvalue $\lambda_{\min}(\Sigma) > 0$.

## 2.2 An Inefficient Algorithm

A first order bound for our problem can be obtained by instantiating the MYGA algorithm of Allen-Zhu et al. (2018) for a set of $\Theta(\frac{T}{K^2 d})^{Kd}$ experts that cover the parameter space of policies of the form (3), which is guaranteed to contain the optimal policy $\pi_T^*$:

**Theorem 2.1.** *Suppose that $0 \le \ell_t(a, X_t) \le 1$ almost surely for all $a \in [K]$. Then, by instantiating MYGA with $\Theta(\frac{T}{K^2 d})^{Kd}$ experts, it obtains the following first-order bound for the adversarial linear contextual bandit problem:*

$$R_T = O\left(K\sqrt{dL_T^* \log T} + K^2 d \log T\right). \tag{4}$$

Although this provides a quick way to see that first-order bounds are possible, the resulting algorithm is completely impractical, because its run-time is proportional to the number of experts, which grows as a large polynomial in $T$. The proof, including a more detailed description of the experts, can be found in Appendix A.

## 3 First- and Second-Order Bounds

In this section we present an algorithm using a novel adaptation of a method developed for the adversarial linear bandit to be suitable for use in the adversarial linear contextual bandit setting. The method proposed is based on a form of continuous exponential weights that has been shown to lead to a first-order bound in the former (Ito et al., 2020). The algorithm allows for optimistic estimates $m_{t,a} \in \mathbb{R}^d$ for the environment's choices $\theta_{t,a}$, which can always be set to 0 when they are not available. We show two types of guarantees. First, in Theorem 3.1, we obtain a second-order regret bound in terms of the cumulative squared error of the estimates $m_{t,a}$:

$$V_T = \mathbb{E}\left[\sum_{s=1}^T \langle X_s, \theta_{s,A_s} - m_{s,A_s} \rangle^2\right]. \tag{5}$$

Taking $m_{t,a} = 0$, this provides a second-order regret bound in terms of the squared losses. Alternatively, $m_{t,a}$ may be estimated using an online regression algorithm, as described by Ito et al. (2020). As our second result, we show in Theorem 3.2 that a first-order bound can be derived for the same algorithm with a different choice of hyperparameters and the assumption that the losses are non-negative.

---

**Algorithm 1** CONTEXTEW

**Parameters:** $\gamma > 0, \eta_1 \geq \ldots \geq \eta_T > 0, m_1, \ldots, m_T$

**For** $t = 1, \ldots, T$:

1. Observe $X_t$.

2. **Repeat:**
   Pick $Q_t$ from the distribution $p_t$ defined in (8), **until**

$$\sum_{a=1}^{K} \|Q_{t,a} X_t\|_{\Sigma_{t,a}^{-1}}^2 \leq dK\gamma^2, \tag{6}$$

   where $\Sigma_{t,a}$ is defined in (9).

3. Set $\widetilde{Q}_t = Q_t$ equal to the last sample of $Q_t$, which caused the loop to exit, and choose an arm according to $A_t \sim \widetilde{Q}_t$.

4. Observe the loss $\ell_t(X_t, A_t)$ and estimate $\widehat{\theta}_{t,a}$ for all $a$ according to (12).

---

## 3.1 Algorithm Description

Our full algorithm is shown in Algorithm 1. As it is an adaptation of continuous exponential weights for the contextual bandits setting, we refer to it as CONTEXTEW. It runs a two-stage sampling procedure: after observing context $X_t$, the first stage of the algorithm samples a random policy $\widetilde{Q}_t \in \Delta^K$, and then the second stage consists of drawing an arm $A_t$ randomly from $\widetilde{Q}_t$. The distribution of $\widetilde{Q}_t$ is constructed as follows: first we sample a different policy $Q_t$ from the exponential weights distribution over the probability simplex with density proportional to

$$w_t(q|X_t) = \exp(-\eta_t \sum_{a=1}^{K} q_a \langle X_t, \sum_{s=1}^{t-1} \widehat{\theta}_{s,a} + m_{t,a} \rangle), \tag{7}$$

where $m_{s,a}$ is a function that is measurable with respect to $\mathcal{F}_{s-1}$. The sum $\sum_{a=1}^{K} q_a \left\langle X_t, \sum_{s=1}^{t-1} \widehat{\theta}_{s,a} \right\rangle$ estimates the cumulative loss that the policy $q$ would have incurred if it had been played in all previous rounds. It relies on estimates $\widehat{\theta}_{s,a}$ of the loss vectors $\theta_{s,a}$, which will be defined below, and a time-varying learning rate $\eta_t > 0$, which is hyperparameter of the algorithm. The normalized density function corresponding to the weights in (7) is:

$$p_t(q|X_t) = \frac{w_t(q|X_t)}{\int_{\Delta^K} w_t(q|X_t) dq}. \tag{8}$$

Following Ito et al. (2020), we then introduce a rejection sampling step (6) to reduce the variance, which is based on the following covariance matrices $\Sigma_{t,a}$ corresponding to $Q_t$:

$$\Sigma_{t,a} = \mathbb{E}_t \left[ Q_{t,a}^2 X_t X_t^\intercal \right], \tag{9}$$

so that $\widetilde{Q}_t$ ends up being sampled according to the following truncated exponential weights density:

$$\tilde{p}_t(q|X_t) = \frac{p_t(q|X_t) \mathbb{1} \left\{ \sum_{a=1}^{K} \|q_a X_t\|_{\Sigma_{t,a}^{-1}}^2 \leq dK\gamma^2 \right\}}{\mathbb{P}_t \left[ \sum_{a=1}^{K} \|q_a X_t\|_{\Sigma_{t,a}^{-1}}^2 \leq dK\gamma^2 | X_t \right]}, \tag{10}$$

with truncation level hyperparameter $\gamma > 0$. We will show that all $\Sigma_{t,a}$ are invertible, as are their analogues in which $Q_t$ is replaced by $\widetilde{Q}_t$:

$$\widetilde{\Sigma}_{t,a} = \mathbb{E}_t \left[ \widetilde{Q}_{t,a}^2 X_t X_t^\intercal \right]. \tag{11}$$

It remains to specify our estimators for $\theta_{t,a}$, which are defined as follows:

$$\widehat{\theta}_{t,a} = m_{t,a} + \widetilde{Q}_{t,a} \widetilde{\Sigma}_{t,a}^{-1} X_t \left( \langle X_t, \theta_{t,a} \rangle - \langle X_t, m_{t,a} \rangle \right) \mathbb{1} \{A_t = a\}. \tag{12}$$

These estimates can be shown to be unbiased:

$$\mathbb{E}_t\left[\widehat{\theta}_{t,a}\right] = m_{t,a} + \widetilde{\Sigma}_{t,a}^{-1}\mathbb{E}_t\left[\widetilde{Q}_{t,a}X_tX_t^\intercal \mathbb{1}\left\{A_t = a\right\}\right](\theta_{t,a} - m_{t,a})$$

$$= m_{t,a} + \widetilde{\Sigma}_{t,a}^{-1}\mathbb{E}_t\left[\widetilde{Q}_{t,a}^2X_tX_t^\intercal\right](\theta_{t,a} - m_{t,a}) = \theta_{t,a}.$$

## 3.2 Results

We instantiate CONTEXTEW with adaptive learning rates $\eta_t$. For our second-order result, these are defined in terms of the empirical counterpart to $V_t$: $\widehat{V}_t = \sum_{s=1}^t \langle X_s, \theta_{s,A_s} - m_{s,A_s}\rangle^2$, and we abbreviate $G_t = 8\sqrt{\widehat{V}_{t-1}\ln(2T^2) + 144\ln^2 T} + 176\ln T$. Then we set

$$\eta_t = (100dK\gamma^2 + d(\widehat{V}_{t-1} + 1 + G_{t-1}))^{-1/2}. \tag{13}$$

This leads to the following second-order bound:

**Theorem 3.1** (Second-Order). *Suppose $\mathcal{D}$ has a log-concave density. Then, for $\gamma = 4\log(10dKT)$, $\eta_t$ as in (13) and any $\mathcal{F}_{t-1}$-measurable estimates $m_t$, the expected regret of* CONTEXTEW *is at most $R_T = \widetilde{O}(K\sqrt{dV_T})$.*

To tune $\eta_t$ adaptively for our first-order bound, we define it using the algorithm's empirical cumulative loss $\widehat{L}_t = \sum_{s=1}^t \ell_t(X_s, A_s)$, which acts as a self-confident empirical estimate of $L_T^*$. We further abbreviate

$$H_t = 8\sqrt{2\widehat{L}_t\ln T + 40\ln^2 T} + 72\ln T, \tag{14}$$

and then set

$$\eta_t = (100d\gamma^2 + dK(\widehat{L}_{t-1} + 1 + H_{t-1}))^{-1/2}. \tag{15}$$

This leads to the following first-order bound:

**Theorem 3.2** (First-Order). *Suppose that $\mathcal{D}$ has a log-concave density and that $0 \le \ell_t(a, X_t) \le 1$ almost surely for all $a \in [K]$. Then, for $\gamma = 4\log(10dKT)$, $\eta_t$ as in (15) and $m_t = 0$, the expected regret of* CONTEXTEW *is at most $R_T = \widetilde{O}(K\sqrt{dL_T^*})$.*

## 3.3 Computational Efficiency

The two computational bottlenecks in the algorithm are the cost of sampling from the output distribution $p_t(q|X_t)$ and computation of the covariance matrices $\Sigma_{t,a}$ in each round.

Due to the log-linearity of our method, there exists several practical methods of sampling. As mentioned in Ito et al. (2020), one can employ the methods of Lovász and Vempala (2007), which was shown in Lovász and Vempala (2006) to enjoy a bound of $O(K^4\log(1/\epsilon))$ (where $\epsilon$ is a bound on the total variation distance between the output distribution and the target), but this still requires knowledge of a density dominating the target distribution on all but a set with total starting mass $\le \epsilon/2$. In Narayanan and Rakhlin (2017), a method is developed for general log-concave distributions which, specialized to log-linear distributions (and without additional assumptions on the initial distribution) yields an $O(K^3\nu^2 + \log(1/\epsilon))$ method when the geometry admits a $\nu$-self concordant barrier. Since there always exists a $K$-self-concordant barrier for a $K$-dimensional convex body, and thus the running time of this method for our problem is $O(K^5 + \log T)$ up to a precision $\epsilon \sim \frac{1}{T^\beta}$ for some $\beta > 0$. As referred to in Ito et al. (2020), the covariance matrix $\Sigma_{t,a}$ is computable in $\mathcal{O}((d/\epsilon)^{O(1)})$ sampling steps drawing upon the results of Lovász and Vempala (2007).

## 4 Analysis

In this section we provide the analysis of CONTEXTEW from which Theorems 3.1 and 3.2 follow. Throughout the analysis, we will be extensively using the following property of log-concave distributions:

**Lemma 4.1.** *If $x$ follows a log-concave distribution $p$ over $\mathbb{R}^d$ and $\mathbb{E}[xx^\intercal] \preccurlyeq I$, we have, for any $\alpha \ge 0$:*

$$\mathbb{P}\left[\|x\|_2^2 \ge d\alpha^2\right] \le d\exp(1 - \alpha). \tag{16}$$

This result was proven in Lemma 1 in Ito et al. (2020), and also follows from Lemma 5.7 in Lovász and Vempala (2007).

First, we need to introduce some notation which will be useful for the reduction to the linear bandit setting and for the accompanying proofs. We denote $z_a(q,x) = q_a x$ and $z(q,x) = (z_1(q,x), \ldots, z_K(q,x))^\intercal$. We also define $\Sigma_t = \operatorname{diag}_{a \in [K]}(\Sigma_{a,t})$ as a block diagonal arrangement of the covariance matrices per arm. Using this notation, the distribution of the sampling algorithm (10) may be rewritten as

$$\tilde{p}_t(q|x) = \frac{p_t(q|x) \mathbb{1}\left\{\|z(q,x)\|^2_{\Sigma_t^{-1}} \le dK\gamma^2\right\}}{\mathbb{P}_t\left[\|z(q,x)\|^2_{\Sigma_t^{-1}} \le dK\gamma^2\right]}. \tag{17}$$

Let $\widetilde{Q}_t(x) \sim \tilde{p}_t(q|x)$, $Q_t(x) \sim p_t(q|x)$ and $\tilde{Z}_t(x) = z(\widetilde{Q}_t(x),x)$, $Z_t(x) = z(Q_t(x),x)$, $Z^*(x) = z(\pi^*(x),x)$. And we denote the aggregated loss parameter $\theta_t = (\theta_1, \ldots, \theta_K)^\intercal$ and its estimate $\widehat{\theta}_t = (\widehat{\theta}_1, \ldots, \widehat{\theta}_K)^\intercal$. Then we can express the regret as follows:

$$R_T = \mathbb{E}\left[\sum_{t=1}^T \ell_t(X_t, A_t) - \ell_t(X_t, \pi^*(X_t))\right] = \mathbb{E}\left[\sum_{t=1}^T \left\langle \tilde{Z}_t(X_t) - Z^*(X_t), \theta_t \right\rangle\right]. \tag{18}$$

The crucial observation is that the log-concavity of the distribution of $Z_t(X_t)$ follows from that of the distribution of $X_t$:

**Lemma 4.2.** *Suppose $z(q,x) = \sum_a q_a \varphi(x,a)$ for $\varphi(x,a) = (\bar{0}^\intercal, \ldots, x^\intercal, \cdots)$ such that $x$ is on the $da$'th co-ordinate and $Q(x) \sim p_t(\cdot|x)$ for $p_t(\cdot|x)$ defined in (8). If $X \sim p_X(\cdot)$ and $p_X(\cdot)$ is log-concave and $Z(x) = z(Q_t(x),x)$, then $Z(X)$ also follows a log-concave distribution.*

The proof of this result is a rather straightforward computation of the density of $Z_t(X_t)$ and can be found in Appendix C. To proceed, we write regret as a sum of two terms

$$R_t = \mathbb{E}\left[\sum_{t=1}^T \left\langle \widetilde{Z}_t(X_t) - Z_t(X_t), \theta_t \right\rangle\right] + \mathbb{E}\left[\sum_{t=1}^T \left\langle Z_t(X_t) - Z^*(X_t), \theta_t \right\rangle\right]. \tag{19}$$

Having shown that $Z_t(X_t)$ is log-concave, and since the log-concavity is preserved under linear transformations, for $y = \Sigma_t^{-1/2\intercal} Z_t(X_t)$ we can see that $\mathbb{E}\left[yy^\intercal\right] = I$, and thus by Lemma 4.1 it immediately follows that the probability that (6) is not satisfied is small for a proposed choice of $\gamma = 4\log(10dKT)$:

$$\mathbb{P}_t\left[\|Z_t(X_t)\|^2_{\Sigma_t^{-1}} > dK\gamma^2\right] \le dK\exp(1-\gamma) \le 3dK\exp(-\gamma) \le \frac{1}{6T^2}.$$

Using this observation, we show that the first term of (19) is just $\mathcal{O}(1)$, which is formally proved in Lemma C.2 in the appendix.

To control the second term of the regret decomposition (19), consider the reduction of the contextual bandit problem to a combination of auxiliary online learning problems that are defined separately for each context, as proposed in Neu and Olkhovskaya (2020), Lemma 3. More details and a full proof can be found in Appendix C.

**Lemma 4.3.** *Let $\pi^*$ be any fixed stochastic policy and let $X_0 \sim \mathcal{D}$ be a sample from the context distribution independent from $\mathcal{F}_T$. Suppose that $p_t \in \mathcal{F}_{t-1}$, such that $p_t(\cdot|x)$ is a probability density with respect to Lebesgue measure with support $\Delta^K$ and let $Q_t(x) \sim p_t(\cdot|x)$. Then,*

$$\mathbb{E}_t\left[\langle Z_t(X_t) - Z^*(X_t), \theta_t \rangle\right] = \mathbb{E}_t\left[\left\langle Z_t(X_0) - Z^*(X_0), \widehat{\theta}_t \right\rangle\right]. \tag{20}$$

To see why this would be useful further in the proof, we interpret the right-hand side of (20) as follows. Consider the online learning problem for a fixed $x$ with the decision set to be $\Delta^K$ and losses $\ell_t(x,q) = \langle z(q,x), \widehat{\theta}_t \rangle$ and consider running a version of a contextual bandit problem with a fixed context $x$, such that the probability of an action $q$ defined as in Equation 8, so $p_t(q|x) \propto \exp\left(-\eta_t \sum_{a=1}^K q_a \left\langle x, \sum_{s=1}^{t-1} \widehat{\theta}_{s,a} \right\rangle\right)$. Then, the regret for the fixed $x$ against $\pi^*(x)$ can be written as:

$$\widehat{R}_T(x) = \sum_{t=1}^T \mathbb{E}_{Q_t(x) \sim p_t(\cdot|x)}\left[\langle z(Q_t(x),x) - z(\pi^*(x),x), \widehat{\theta}_t \rangle\right].$$

Then it is easy to see that the right-hand side of (20) is equal to $\mathbb{E}\left[\widehat{R}_T(X_0)\right]$. Thus, we first show a bound on $\widehat{R}_T(x)$ that holds almost surely for any $x$ and then take an expectation with respect to $X_0$. We control the regret $\widehat{R}_T(x)$ by following the general schema of the optimistic mirror descent analysis developed in (Rakhlin and Sridharan, 2013; Ito et al., 2020). With this analysis, we get the following bound for any $x \in \mathcal{X}$:

**Lemma 4.4.** *Assume that $\eta_{t+1} \leq \eta_t$ for all $t$, let $q_0$ be a uniform distribution over $[K]$ and $\psi(y) = \exp(y) - y - 1$. Then, the regret $\widehat{R}_T(x)$ of* CONTEXTEW *almost surely satisfies*

$$\widehat{R}_T(x) \leq \frac{1}{T}\sum_{t=1}^{T}\left\langle z(q_0 - \pi^*(x), x), \widehat{\theta}_t\right\rangle + \frac{K\log T}{\eta_T}$$
$$+ \sum_{t=1}^{T}\frac{1}{\eta_t}\mathbb{E}_{Q_t(x)\sim p_t(\cdot|x)}\left[\psi\left(-\eta_t\left\langle z(Q_t(x), x), \widehat{\theta}_t - m_t\right\rangle\right)\right], \tag{21}$$

*for $\psi(y) = \exp(y) - y - 1$.*

We place the derivation of the this bound in the appendix. The crucial ingredient is to show that the square of the estimated loss can be bounded by the square of the true loss. Using the definition of $\theta_t$, denoting $Var_t = \text{tr}\left(\widetilde{\Sigma}_t^{-1}Z_t(X_0)Z_t(X_0)^{\mathsf{T}}\widetilde{\Sigma}_t^{-1}Z_t(X_t)Z_t(X_t)^{\mathsf{T}}\right)$, we get

$$\mathbb{E}_t\left[\left(-\eta_t\left\langle Z_t(X_0), \widehat{\theta}_t - m_t\right\rangle\right)^2\right] = \mathbb{E}_t\left[\eta_t^2\left(\ell_t(A_t, X_t) - X_t^{\mathsf{T}}m_{t,A_t}\right)^2 Var_t\right], \tag{22}$$

As additional corollary of the concentration result for log-concave random variables, we can show the following relation between matrices $\Sigma_t$ and $\widetilde{\Sigma}_t$:

$$\frac{3}{4}\Sigma_t \preceq \widetilde{\Sigma}_t \preceq \frac{4}{3}\Sigma_t, \tag{23}$$

which we prove in Lemma C.2 in the appendix. Then we can show that, almost surely:

$$\mathbb{E}_{X_0}[Var_t] = \mathbb{E}_{X_0}\left[\text{tr}\left(\widetilde{\Sigma}_t^{-1}Z_t(X_0)Z_t(X_0)^{\mathsf{T}}\widetilde{\Sigma}_t^{-1}Z_t(X_t)Z_t(X_t)^{\mathsf{T}}\right)\right]$$
$$= \text{tr}\left(\widetilde{\Sigma}_t^{-1}\Sigma_t\widetilde{\Sigma}_t^{-1}Z_t(X_t)Z_t(X_t)^{\mathsf{T}}\right) \leq \frac{4}{3}\text{tr}\left(\widetilde{\Sigma}_t^{-1}\widetilde{\Sigma}_t\widetilde{\Sigma}_t^{-1}Z_t(X_t)Z_t(X_t)^{\mathsf{T}}\right)$$
$$= \frac{4}{3}Z_t(X_t)^{\mathsf{T}}\widetilde{\Sigma}_t^{-1}Z_t(X_t) \leq Z_t(X_t)^{\mathsf{T}}\Sigma_t^{-1}Z_t(X_t) \leq dK\gamma^2. \tag{24}$$

where the first inequality follows from (23) and the second inequality is immediate from (23) and the fact that for symmetric positive definite matrices $A \succeq B$ follows from $B^{-1} \succeq A^{-1}$. The last inequality follows from (6) in the CONTEXTEW. So, from (22) and (35), we get

$$\mathbb{E}_t\left[\left(-\eta_t\left\langle Z_t(X_0), \widehat{\theta}_t - m_t\right\rangle\right)^2\right] \leq dK\gamma^2\mathbb{E}_t\left[\eta_t^2\left(\ell_t(A_t, X_t) - X_t^{\mathsf{T}}m_{t,A_t}\right)^2\right],$$

which, as we stated above, is the key step to prove Theorem 3.1.

**First-order regret bound** To prove result of Theorem 3.2, we show that the bound in the Theorem 3.1 can instantiated to obtain a first-order regret bound with a different choice of the learning rate $\eta_t$. Going along the same lines with regard to the concentration of $\widehat{L}_t$ as for $\widehat{V}_t$, by setting $m_t = \bar{0}$ and noticing that then $V_T \leq L_T$ we get

$$R_T \leq 2dK\gamma^2\mathbb{E}\left[\sum_{t=1}^{T}\eta_t\ell_t(A_t, X_t)^2\right] + \widetilde{\mathcal{O}}(K\sqrt{dV_T}) \leq 4\sqrt{d}K\gamma^2\sqrt{L_T} + \widetilde{\mathcal{O}}(K\sqrt{dL_T}).$$

Since $R_T = L_t - L_T^*$, by solving the quadratic inequality with respect to $L_T^*$, we get that $L_T \leq L_T^* + \widetilde{\mathcal{O}}(K\sqrt{d})$, yielding the final bound.

# 5 Discussion

In conclusion, by applying the approach of (Ito et al., 2020) we have constructed the first scheme achieving $\tilde{O}\left(K\sqrt{dL_T^*}\right)$ regret with a runtime of $\mathcal{O}\left(\left(K^5 + \log T\right) \cdot g_\Sigma\right)$, where $g_\Sigma$ is the time taken to construct the covariance matrix per round - a potentially large polynomial improvement over the $\mathcal{O}\left(T^{Kd}\right)$ runtime of MYGA. The application of linear bandit algorithms to the contextual bandit problem constitutes, to the best of our knowledge, a novel approach. In doing so we've found a number of positive aspects, including efficiency, but also the direct applicability of other properties enjoyed by the algorithm such as second order bounds (Ito et al., 2020).

Our approach is based on reducing the linear contextual bandit problem to a linear bandit problem, as opposed to a multi-armed bandit problem as in (Neu and Olkhovskaya, 2020). While the specifics of this reduction heavily relied on the joint log-concavity of the context distributions and the exponential-weights posterior over the simplex of actions, we wonder if such approaches can be successfully applied to achieve other types of improvements for linear contextual bandits. In particular, it is curious to what extent other recent advances in the linear bandit problem can be translated to the linear contextual bandit setting. Note that, while the truncation step in Algorithm 1 has an insignificant computational cost as the condition is satisfied with probability $\mathcal{O}(1 - 1/T)$, it can be removed by paying a $\log(1/\lambda_{min}(\Sigma))$ multiplicative term in the regret by implementing additional exploration with probability $1/T$. It is natural to ask whether or not approaches based on other instantiations of online mirror descent would also yield first-order bounds, and possibly improve the dependence on $K$. The answer is not obvious: for an example of how a naive application of an instantiation of FTRL fails to achieve a first-order bound, see Appendix D.

A relevant question pertains to whether or not such an application of algorithms for linear bandits is necessary at all, but standard approaches such as direct adaptation of Exp3, and first-order adaptations thereof such as GREEN Allenberg et al. (2006) do not seem to give the desired result.In addition, thresholding the worst performing arms inevitably biases the loss estimator due to undersampling of those arms for which the threshold has been applied, and the resulting additional bias term picked up in the regret scales with $1/\lambda_{\min}(\Sigma_{t,a})$, which may be arbitrarily large. Another standard approach of finding an optimistic estimator yielded no fruit during the course of this study due to the lack of the existence of such an estimator without saving all previous losses explicitly.

Our algorithm achieves the regret bound $\mathcal{O}(K\sqrt{dV_T})$, while the worst case guarantee of LINEXP3 of Neu and Olkhovskaya (2020) is $\mathcal{O}(\sqrt{dKT})$. This discrepancy is not surprising as the Algorithm 1 of Ito et al. (2020) scales as $\mathcal{O}(n\sqrt{T})$ ($n$ being the dimension of the action space for the linear bandit), which arises from the deployment of continuous exponential weights. MYGA achieves the same $\mathcal{O}(K\sqrt{dL_T^*})$ bound due to the number of experts needed to cover the joint set of additive loss parameters. It is worth here emphasising that no known algorithm achieves a better dependence on $K$ than $\mathcal{O}(K\sqrt{dL_T^*})$ for the linear adversarial contextual bandit problem. Meanwhile, if the linear bandit is played on the $n$-simplex, an improvement to $\sqrt{nT}$ is possible. For further discussion of this point, see Section 28.5 of Lattimore and Szepesvári (2020). It is thus still unclear whether or not the extra factor of $\sqrt{K}$ is necessary if one aims for a first-order bound.

An additional point is that while the MYGA algorithm Allen-Zhu et al. (2018) allows for adversarially chosen contexts, the analysis of MYGA for our setting relies heavily on the assumption that contexts are drawn i.i.d. at each iteration. A natural question is then whether or not a similar result is achievable in the adversarial context case. It is known that achieving sub-linear regret is not possible even for full-information online learning of one-dimensional threshold classifiers when both contexts and losses are adversarial (Ben-David et al., 2009; Syrgkanis et al., 2016), which renders sub-linear regret similarly impossible to guarantee for the even harder setting that we consider in this paper. However, we do conjecture that we could overcome the assumption that the distribution is known or that we can sample from it by employing a more elaborate algorithm to estimate the distribution from the data. Indeed, it is not obvious if the distributional assumption of a lower bound to the covariance matrix eigenvalues is entirely necessary, since the regret does not depend on this.

Lastly, it would be an interesting challenge to see if a high-probability regret bound could be obtained in the form stated in the COLT 2017 open problem Agarwal et al. (2017) for this setting, but since a high-probability $O(\sqrt{T})$ has not yet been proved for the problem here considered, the latter may be more worthy of focus in the short term.

# 6 Acknowledgments

Tim van Erven and Jack Mayo were supported by the Netherlands Organization for Scientific Research (NWO) under grant number VI.Vidi.192.095. Gergely Neu was supported by the European Research Council (ERC) under the European Union's Horizon 2020 research and innovation programme (Grant agreement No. 950180). Chen-Yu Wei would like to acknowledge the support from Simons-Berkeley Research Fellowship.

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

# A First-order Bound by Reduction to MYGA

*Proof.* The MYGA algorithm of Allen-Zhu et al. (2018) competes with a class of experts $E$, where each expert $e \in E$ provides a stochastic prediction $\xi_t^e \in \Delta_K$ in each round $t$. It provides the following expected regret bound with respect to the best expert:

$$R_T = O\left(\sqrt{K \log(|E| + T) L_T^*} + K \log(|E| + T)\right). \tag{25}$$

Losses for the arms can be adversarial, and are assumed to take values in $[0, 1]$.

We will instantiate the experts to cover the parameter space $\{\beta \in \mathbb{R}^{K \times d} : \max_a \|\beta_a\| \leq RT\}$ of potentially optimal parameters for deterministic policies of the form (3), which we know must contain the optimal policy $\pi_T^*$ with corresponding parameters $\beta^* = \mathbb{E}[\sum_{t=1}^T \theta_t]$. The covering number for a ball of radius $RT$ at precision $\epsilon > 0$ is between $\left(\frac{RT}{\epsilon}\right)^d$ and $\left(\frac{3RT}{\epsilon}\right)^d$, so by taking the Cartesian product of this covering with itself $K$ times we can cover all $\beta$ with $\left(\frac{RT}{\epsilon}\right)^{Kd} \leq |E| \leq \left(\frac{3RT}{\epsilon}\right)^{Kd}$ points $\beta^1, \ldots, \beta^{|E|}$. Let $\ddot{\beta} \in \{\beta^1, \ldots, \beta^{|E|}\}$ be the closest point in the covering to the optimal parameters $\beta^*$. Then its expected approximation error can be upper bounded as follows:

$$
\begin{aligned}
\mathbb{E}\left[\sum_{t=1}^T \left\langle X_t, \theta_{t, \pi_{\ddot{\beta}}(X_t)} \right\rangle - \left\langle X_t, \theta_{t, \pi_{\beta^*}(X_t)} \right\rangle\right] &= \mathbb{E}\left[\sum_{t=1}^T \left\langle X_0, \theta_{t, \pi_{\ddot{\beta}}(X_0)} \right\rangle - \left\langle X_0, \theta_{t, \pi_{\beta^*}(X_0)} \right\rangle\right] \\
&= \mathbb{E}\left[\left\langle X_0, \beta^*_{\pi_{\ddot{\beta}}(X_0)} \right\rangle - \left\langle X_0, \beta^*_{\pi_{\beta^*}(X_0)} \right\rangle\right] \\
&\leq \mathbb{E}\left[\left\langle X_0, \ddot{\beta}_{\pi_{\ddot{\beta}}(X_0)} \right\rangle - \left\langle X_0, \beta^*_{\pi_{\beta^*}(X_0)} \right\rangle\right] + \sigma\epsilon \\
&= \mathbb{E}\left[\min_a \left\langle X_0, \ddot{\beta}_a \right\rangle - \left\langle X_0, \beta^*_{\pi_{\beta^*}(X_0)} \right\rangle\right] + \sigma\epsilon \\
&\leq \mathbb{E}\left[\left\langle X_0, \ddot{\beta}_{\pi_{\beta^*}(x_0)} \right\rangle - \left\langle X_0, \beta^*_{\pi_{\beta^*}(x_0)} \right\rangle\right] + \sigma\epsilon \\
&\leq \mathbb{E}\left[\left\langle X_0, \beta^*_{\pi_{\beta^*}(x_0)} \right\rangle - \left\langle X_0, \beta^*_{\pi_{\beta^*}(x_0)} \right\rangle\right] + 2\sigma\epsilon \\
&= 2\sigma\epsilon.
\end{aligned}
$$

Adding this to (25), instantiated with $|E| \leq \left(\frac{3RT}{\epsilon}\right)^{Kd}$, and choosing $\epsilon = \frac{dK^2}{2}$ completes the proof. $\qquad \square$

# B Auxiliary lemmas

To ensure that step 2 in CONTEXTEW is defined correctly, we show that the matrix $\Sigma_t$ is full rank:

**Lemma B.1.** *Let the distribution of $X_t$ be such that $\lambda_{\min}(\mathbb{E}[X_t X_t^\intercal]) > 0$. Then, we can show*

$$\lambda_{\min}(\Sigma_{t,a}) > 0 \tag{26}$$

*for any $a \in [K]$, and consequently*

$$\lambda_{\min}(\Sigma_t) > 0 \tag{27}$$

*Proof.* To show that $\Sigma_{t,a}$ is full rank, it suffices to show that there is no $v \in \mathbb{R}^d$ such that $v^\intercal \Sigma_{t,a} v = 0$. Suppose, to the contrary, that such a $v$ does exist. Then $0 = v^\intercal \mathbb{E}_t\left[Q_{t,a}^2(X_t) X_t X_t^\intercal\right] v = \mathbb{E}_t\left[Q_{t,a}^2(v^\intercal X_t)^2\right]$, which implies that $Q_{t,a} v^\intercal X_t = 0$ almost surely. Since $Q_{t,a} > 0$ almost surely, it follows that in fact $v^\intercal X_t = 0$ almost surely and therefore $0 = \mathbb{E}_t\left[(v^\intercal X_t)^2\right] = v^\intercal \mathbb{E}_t[X_t X_t^\intercal] v$. But this contradicts our assumption that $\lambda_{\min}(\mathbb{E}_t[X_t X_t^\intercal]) > 0$. $\qquad \square$

We will use a simple corollary of Freedman's inequality Freedman (1975) that was introduced in Lemma 2 in Bartlett et al. (2008):

**Lemma B.2.** *Let $Y_1, \ldots, Y_t$ be a martingale difference sequence with respect to a filtration $\mathcal{F}_1 \subset \cdots \subset \mathcal{F}_t$ such that $\mathbb{E}[Y_s | \mathcal{F}_s] = 0$. Suppose that $Y_s \leq b$ holds almost surely. Then with probability at least $1 - \delta$ we have $\sum_{s=1}^t Y_s \leq 2 \max\{2\sqrt{\sum_{s=1}^t \mathbb{E}[Y_s^2 | \mathcal{F}_s]}, b\sqrt{\ln(1/\delta)}\}\sqrt{\ln(1/\delta)}$.*

In our analysis, we will use some results from Ito et al. (2020). We will make use of the following concentration property of log-concave distributions, Lemma 1 from Ito et al. (2020):

**Lemma B.3.** *If $y$ follows a log-concave distribution $p$ over $\mathbb{R}^d$ and $\mathbb{E}_{y \sim p}[yy^\intercal] \preccurlyeq I$, we have*

$$\mathbb{P}\left[\|y\|_2^2 \geq d\alpha^2\right] \leq d\exp(1-\alpha),$$

*for arbitrary $\alpha \geq 0$.*

The lemma presented below introduces a property that will be instrumental in analyzing the variance of loss estimates, presented as Lemma 6 in Ito et al. (2020):

**Lemma B.4.** *If $y$ follows a log-concave distribution over $\mathbb{R}$, and if $\mathbb{E}\left[y^2\right] \leq 1/100$, we have*

$$\mathbb{E}\left[\psi(y)\right] \leq \mathbb{E}\left[y^2\right] + 30\exp\left(-\frac{1}{\sqrt{\mathbb{E}\left[y^2\right]}}\right) \leq 2\mathbb{E}\left[y^2\right], \; \text{where } \psi(x) = \exp(x) - x - 1.$$

## C Proof of Theorem 3.1

The proof of Theorem 3.1 proceeds in a sequence of lemmas. First, we need to show that the distribution of $Z_t(X_t)$ is log-concave for all $t \in [T]$, and after we follow the analysis of Algorithm 1 of Ito et al. (2020), bounding both components of (19) taking into account the required alterations to incorporate contextual structure

**Lemma C.1.** *Suppose $z(q,x) = \sum_a q_a \varphi(x,a)$ for $\varphi(x,a) = (\bar{0}^\intercal, \ldots, x^\intercal, \cdots)$ such that $x$ is on the $da$'th co-ordinate and $Q(x) \sim p(\cdot|x)$ for $p(\cdot|x)$ log-concave. If $X \sim p_X(\cdot)$ and $p_X(\cdot)$ is log-concave and $Z(x) = z(Q(x), x)$, then $Z(X)$ also follows a log-concave distribution.*

*Proof.* Assume that $|x^i| > 0$ for all $i \in [d]$. Set $(z_1, \ldots, z_{K-1}, x) = h(q_1\bar{1}, \ldots, q_{K-1}\bar{1}, x)$, where $h : \mathbb{R}^{dK} \to \mathbb{R}^{dK}$ and $z_i = (\ldots, z_i^j, \ldots)^\intercal$ for each $i \in \{1, \ldots, K-1\}$. Thus $z_i^j = h_i(q_i, x^j) = q_i(x^j)$ and $h_K(x) = x$. The Jacobian of $h^{-1}(z_1, \ldots, z_{K-1}, x)$ can be expressed as the block matrix

$$J(h^{-1}(z_1, \ldots, z_{K-1}, x)) = \begin{bmatrix} \Lambda_x & \Gamma_{z,x} \\ 0_{d \times (K-1)} & \mathrm{Id}_{d \times d} \end{bmatrix},$$

where $\Lambda_x \in \mathbb{R}^{(K-1) \times (K-1)}$ is diagonal with $(\Lambda_x)_{ii} = \frac{1}{x^i}$ and $\Gamma_{z,x} \in \mathbb{R}^{(K-1) \times d}$ with $(\Gamma_{z,x})_{ij} = -\frac{z_i^j}{(x^j)^2}$. Since $J(h^{-1}(z_1, \ldots, z_{K-1}, x))$ is upper-triangular, $\det(J(h^{-1}(z_1, \ldots, z_{K-1}, x))) = \left(\prod_{i=1}^d \frac{1}{x^i}\right)^{K-1}$. The joint distribution of $Z_i$ and $X$ can thus be written

$$p_{Z_1, \ldots, Z_{K-1}, X}(z_1, \ldots, z_{K-1}, x) = p_{Q,X}\left(h^{-1}(z_1, \ldots, z_{K-1}, x)\right)\left(\prod_{i=1}^d \frac{1}{x^i}\right)^{K-1}$$

with the joint distribution between $Q$ and $X$ of the form

$$p_{Q,X}\left(h^{-1}(z_1, \ldots, z_{K-1}, x)\right) = \frac{e^{-\eta\langle\psi(z,x,\varphi),\widehat{\Theta}_{t-1}\rangle}}{\int_{q' \in C} e^{-\eta\langle\sum_{a=1}^{K-1} q_a'\varphi(x,a),\widehat{\Theta}_{t-1}\rangle}dq'}p_X(x)$$

where $(\psi(z,x,\varphi))_i = \sum_{a=1}^{K-1}\frac{z_a^i}{x^i}\varphi(x,a)^i + \left(1 - \sum_{a=1}^{K-1}\frac{z_a^i}{x^i}\right)\varphi(x,K)^i$ has been defined for readability. We can reabsorb the factor $\left(\prod_{i=1}^d \frac{1}{x^i}\right)^{K-1}$ in the denominator to rewrite the normalization constant as a in terms of the random variable $Z(x)$, and so

$$p_{Z_1, \ldots, Z_{K-1}, X}(z_1, \ldots, z_{K-1}, x) = \frac{e^{-\eta\langle\psi(z,x,\varphi),\widehat{\Theta}_{t-1}\rangle}}{\int_{z' \in Z(x)} e^{-\eta\langle\psi(z',x,\varphi),\widehat{\Theta}_{t-1}\rangle}dz'}p_X(x).$$

Define a new function $g : \mathbb{R}^{d \times K} \to \mathbb{R}^{d \times K}$ such that $y = g(z_1, \ldots, z_{K-1}, x) = (\ldots, g_i(z_1, \ldots, z_{K-1}, x), \ldots)^\intercal$, where for $i \in [1, K-1]$, $g_i(z_1, \ldots, z_{K-1}, x) = z_i$ and

$g_K(z_1, \ldots, z_{K-1}, x) = (\ldots, \left(1 - \frac{1}{x^i}\sum_{a=1}^{K-1} z_a^i\right)x^i, \ldots)$. Then for $i \in \{1, \ldots, K-1\}$, $g_i^{-1}(y) = y_i$ and $g_K^{-1}(y) = \sum_{a=1}^{K} y_a$. The determinant $\det(J(g^{-1}(y))) = 1$, so

$$p_Y(y) = p_{Z_1, \cdot, Z_{K-1}, X}(g^{-1}(y))$$

$$= \frac{e^{-\eta\langle y, \widehat{\Theta}_{t-1}\rangle}}{\int_{y' \in Y(y)} e^{-\eta\langle y', \widehat{\Theta}_{t-1}\rangle} dy'} p_X\left(\sum_{a=1}^{K} y_a\right).$$

Since both $p_X$ and $\dfrac{e^{-\eta\langle y, \widehat{\Theta}_{t-1}\rangle}}{\int_{y' \in Y(y)} e^{-\eta\langle y', \widehat{\Theta}_{t-1}\rangle} dy'}$ are both log-concave, the lemma follows. $\square$

Having shown the log-concavity of $Z(X_t)$, we may safely proceed.

We state the analog of Lemma 4 in Ito et al. (2020) adapted to our setting, leading to a bound on the first term of (19) as well as providing a useful relation between $\Sigma_t$ and $\widetilde{\Sigma}_t$.

**Lemma C.2.**
$$\left|\mathbb{E}_t\left[\left\langle Z_t(X_t) - \widetilde{Z}_t(X_t), \theta_t\right\rangle\right]\right| \leq \frac{1}{2T^2}, \tag{28}$$

*and we have*
$$\frac{3}{4}\Sigma_t \preceq \widetilde{\Sigma}_t \preceq \frac{4}{3}\Sigma_t. \tag{29}$$

*Proof.* From definition of $\tilde{p}_t$, for any $x \in \mathcal{X}, \theta \in \Theta$, we have

$$\mathbb{E}_t\left[\left\langle \widetilde{Z}_t(X_t), \theta\right\rangle\right]$$

$$= \frac{1}{\mathbb{P}_t\left[\|Z_t(X_t)\|_{\Sigma_t^{-1}}^2 \leq dK\gamma^2\right]} \int_{\Delta^K} \int_{\mathcal{X}} \langle z(q, x), \theta\rangle \mathbb{1}\left\{\|z(q, x)\|_{\Sigma_t^{-1}}^2 \leq dK\gamma^2\right\} p_t(q|x)p(x)dxdq$$

$$= \frac{1}{1 - \delta} \int_{\Delta^K} \int_{\mathcal{X}} \langle z(q, x), \theta\rangle \mathbb{1}\left\{\|z(q, x)\|_{\Sigma_t^{-1}}^2 \leq dK\gamma^2\right\} p_t(q|x)p(x)dxdq$$

$$= \frac{1}{1 - \delta}\left(\mathbb{E}_t\left[\langle Z_t(X_t), \theta\rangle\right] - \int_{\Delta^K} \int_{\mathcal{X}} \langle z(q, x), \theta\rangle \mathbb{1}\left\{\|z(q, x)\|_{\Sigma_t^{-1}}^2 > dK\gamma^2\right\} p_t(q|x)p(x)dxdq\right),$$

where $\delta = \mathbb{P}_t\left[\|Z_t(X_t)\|_{\Sigma_t^{-1}}^2 > dK\gamma^2\right]$. Plugging this into the l.h.s. of (28) yields

$$\left|\mathbb{E}_t\left[\left\langle Z_t(X_t) - \widetilde{Z}_t(X_t), \theta_t\right\rangle\right]\right|$$

$$= \frac{1}{1-\delta}\left|\delta\mathbb{E}_t\left[\langle Z_t(X_t), \theta_t\rangle\right] + \int_{\Delta^K}\int_{\mathcal{X}} \langle z(q, x), \theta\rangle\mathbb{1}\left\{\|z(q, x)\|_{\Sigma_t^{-1}}^2 > dK\gamma^2\right\} p_t(q|x)p(x)dxdq\right|$$

$$\leq \frac{1}{1-\delta}\left(\delta + \int_{\Delta^K}\int_{\mathcal{X}}\mathbb{1}\left\{\|z(q, x)\|_{\Sigma_t^{-1}}^2 > dK\gamma^2\right\} p_t(q|x)p(x)dxdq\right) = \frac{2\delta}{1-\delta}.$$

Since the distribution of $Z_t(X_t)$ is log-concave (Lemma C.1), we can apply Lemma 1 of Ito et al. (2020) to $x = \Sigma_t^{-1/2}Z_t(X_t)$. The assumptions of Lemma 1 of Ito et al. (2020) hold since we have $\mathbb{E}\left[xx^\mathsf{T}\right] = I$ and since log-concavity is preserved under linear maps. Using Lemma 1 of Ito et al. (2020), we have

$$\delta = \mathbb{P}_t\left[\|Z_t(X_t)\|_{\Sigma_t^{-1}}^2 > dK\gamma^2\right] \leq dK\exp(1 - \gamma) \leq 3dK\exp(-\gamma) \leq \frac{1}{6T^2},$$

where the last inequality follows from $\gamma \geq 4\log(10dKT)$, which obtains (28). We proceed to showing (29). For any $y \in \mathbb{R}^{dK}$, we have

$$y^\mathsf{T}\widetilde{\Sigma}_t y = \mathbb{E}\left[(y^\mathsf{T}\widetilde{Z}_t(X_t))^2\right] = \frac{1}{1-\delta}\mathbb{E}_t\left[(y^\mathsf{T}Z_t(X_t))^2\mathbb{1}\left\{\|Z_t(X_t)\|_{\Sigma_t^{-1}}^2 \leq dK\gamma^2\right\}\right]$$

$$\leq \frac{1}{1-\delta}\mathbb{E}_t\left[(y^\mathsf{T}Z_t(X_t))^2\right] = \frac{1}{1-\delta}y^\mathsf{T}\Sigma_t y.$$

Since this holds for all $y \in \mathbb{R}^{dK}$ and $\frac{1}{1-\delta} \leq \frac{4}{3}$, the second inequality in (29) holds. Furthermore, we have

$$
\begin{aligned}
y^\intercal \Sigma_t y - y^\intercal \widetilde{\Sigma}_t y &= \mathbb{E}_t \left[ (y^\intercal Z_t(X_t))^2 \right] - \frac{1}{1-\delta} \mathbb{E}_t \left[ (y^\intercal Z_t(X_t))^2 \mathbb{1}\left\{ \|Z_t(X_t)\|_{\Sigma_t^{-1}}^2 \leq dK\gamma^2 \right\} \right] \\
&\leq \mathbb{E}_t \left[ (y^\intercal Z_t(X_t))^2 \mathbb{1}\left\{ \|Z_t(X_t)\|_{\Sigma_t^{-1}}^2 > dK\gamma^2 \right\} \right] \\
&\leq y^\intercal \Sigma_t y \, \mathbb{E}_t \left[ \|Z_t(X_t)\|_{\Sigma_t^{-1}}^2 \mathbb{1}\left\{ \|Z_t(X_t)\|_{\Sigma_t^{-1}}^2 > dK\gamma^2 \right\} \right],
\end{aligned}
\tag{30}
$$

where the last inequality follows from Cauchy-Schwarz:

$$
(y^\intercal Z_t(X_t))^2 = \left( \left\langle \Sigma_t^{1/2} y, \Sigma_t^{-1/2} x \right\rangle \right)^2 \leq \left\| \Sigma_t^{1/2} y \right\|_2^2 \cdot \left\| \Sigma_t^{-1/2} x \right\|_2^2 = y^\intercal \Sigma_t y \, \|x\|_{\Sigma_t^{-1}}^2.
$$

The right-hand side of (30) can be bounded using Lemma B.3 as follows:

$$
\begin{aligned}
\mathbb{E}_t &\left[ \|Z_t(X_t)\|_{\Sigma_t^{-1}}^2 \mathbb{1}\left\{ \|Z_t(X_t)\|_{\Sigma_t^{-1}}^2 > dK\gamma^2 \right\} \right] \\
&\leq \sum_{n=1}^\infty (n+1)^2 dK\gamma^2 \mathbb{P}_t \left[ n^2 dK\gamma^2 \leq \|Z_t(X_t)\|_{\Sigma_t^{-1}}^2 \leq (n+1)^2 dK\gamma^2 \right] \\
&\leq \sum_{n=1}^\infty (n+1)^2 (dK)^2 \gamma^2 \exp(1 - n\gamma) \\
&\leq (dK)^2 \gamma^2 \sum_{n=1}^\infty \exp(2 + n - n\gamma) = (dK)^2 \gamma^2 \frac{\exp(3-\gamma)}{1 - \exp(1-\gamma)} \leq \frac{1}{4}.
\end{aligned}
\tag{31}
$$

Combining (31) and (30) we get the first inequality of (29). $\qquad \square$

**Lemma C.3.** *Let $\pi^*$ be any fixed stochastic policy and let $X_0 \sim \mathcal{D}$ be a sample from the context distribution independent from $\mathcal{F}_T$. Suppose that $p_t \in \mathcal{F}_{t-1}$, such that $p_t(\cdot|x)$ is a probability density with respect to Lebesgue measure with support $\Delta^K$ and let $Q_t(x) \sim p_t(\cdot|x)$. Then,*

$$
\mathbb{E} \left[ \sum_{t=1}^T \langle z(Q_t(X_t), X_t) - z(\pi^*(X_t), X_t), \theta_t \rangle \right] = \mathbb{E} \left[ \sum_{t=1}^T \left\langle z(Q_t(X_0), X_0) - z(\pi^*(X_0)), X_0, \widehat{\theta}_t \right\rangle \right].
$$

*Proof.* For any $t$, we have

$$
\begin{aligned}
\mathbb{E}_t \left[ \left\langle Z_t(X_0) - Z^*(X_0), \widehat{\theta}_t \right\rangle \right] &= \mathbb{E}_t \left[ \mathbb{E}_t \left[ \left\langle Z_t(X_0) - Z^*(X_0), \widehat{\theta}_t \right\rangle \Big| X_0 \right] \right] \\
&= \mathbb{E}_t \left[ \mathbb{E}_t \left[ \langle Z_t(X_0) - Z^*(X_0), \theta_t \rangle | X_0 \right] \right] = \mathbb{E}_t \left[ \langle Z_t(X_t) - Z^*(X_t), \theta_t \rangle \right].
\end{aligned}
$$

$\qquad \square$

Then, we prove the almost sure regret bound for any $x$ and then take an expectation over $X_0$. We further proceed with an adaptation of the analysis of the continuous exponential weights algorithm, which was stated in Ito et al. (2020) as Lemma 16, but we include it here for the clarity. Let $\psi(y) = \exp(y) - y - 1$. For any $x \in \mathcal{X}$, we show the following :

**Lemma C.4.** *Assume that $\eta_{t+1} \leq \eta_t$ for all t, let $q_0$ be a uniform distribution over $[K]$ and $\psi(y) = \exp(y) - y - 1$. Then, the regret $\widehat{R}_T(x)$ for any $x \in \mathcal{X}$ of CONTEXTEW almost surely satisfies*

$$
\widehat{R}_T(x) \leq \frac{1}{T} \sum_{t=1}^T \left\langle z(q_0 - \pi^*(x), x), \widehat{\theta}_t \right\rangle + \frac{K \log T}{\eta_T} + \sum_{t=1}^T \frac{1}{\eta_t} \mathbb{E}_{Q_t(x) \sim p_t(\cdot|x)} \left[ \psi \left( -\eta_t \left\langle z(Q_t(x), x), \widehat{\theta}_t - m_t \right\rangle \right) \right].
$$

*Proof.* Note that we can write $\widehat{R}_T(x)$ as

$$
\widehat{R}_T(x) = \sum_{t=1}^T \left( \int_{\Delta^K} p_t(q|x) \left\langle z(q, x), \widehat{\theta}_t \right\rangle dq - \left\langle z(\pi^*(x), x), \sum_{t=1}^T \widehat{\theta}_t \right\rangle \right).
$$

Define $W_t(x) = \int_{\Delta^K} w_t(q|x) dq$, $u_t(q|x) = \exp\left(-\eta_t \sum_a q_a \left\langle x, \sum_{s=1}^t \widehat{\theta}_{s,a} \right\rangle\right)$, $U_t(x) = \int_{\Delta^K} u_t(q|x) dq$ and $v_t(q|x) = \exp\left(-\eta_{t+1} \sum_a q_a \left\langle x, \sum_{s=1}^t \widehat{\theta}_{s,a} \right\rangle\right)$, $V_t(x) = \int_{\Delta^K} v_t(q|x) dq$. We have

$$U_t(x) = \int_{\Delta^K} w_t(q|x) \exp\left(-\eta_t \left\langle z(q,x), \widehat{\theta}_t - m_t \right\rangle\right) dq = W_t(x) \int_{\Delta^K} p_t(q|x) \exp\left(-\eta_t \left\langle z(q,x), \widehat{\theta}_t - m_t \right\rangle\right) dq$$

$$= W_t(x) \int_{\Delta^K} p_t(q|x) \left(1 - \eta_t \left\langle z(q,x), \widehat{\theta}_t - m_t \right\rangle + \psi(-\eta_t \left\langle z(q,x), \widehat{\theta}_t - m_t \right\rangle)\right) dq.$$

Taking the logarithm of both sides, we get

$$\log(U_t(x)) = \log(W_t(x)) + \log\left(\int_{\Delta^K} p_t(q|x) \left(1 - \eta_t \left\langle z(q,x), \widehat{\theta}_t - m_t \right\rangle + \psi(-\eta_t \left\langle z(q,x), \widehat{\theta}_t - m_t \right\rangle)\right) dq\right)$$

$$\leq \log(W_t(x)) + \int_{\Delta^K} p_t(q|x) \left(-\eta_t \left\langle z(q,x), \widehat{\theta}_t - m_t \right\rangle + \psi(-\eta_t \left\langle z(q,x), \widehat{\theta}_t - m_t \right\rangle)\right) dq,$$
(32)

where we used the inequality $\log(1 + x) \leq x$ for $x > -1$.

$$V_{t-1}(x) = \int_{\Delta^K} w_t(q|x) \exp\left(\eta_t \sum_a q_a \left\langle x, m_{t,a} \right\rangle\right) dq = W_t(x) \int_{\Delta^K} p_t(q|x) \exp\left(\eta_t \sum_a q_a \left\langle x, m_{t,a} \right\rangle\right) dq$$

$$\geq W_t(x) \exp\left(\eta_t \int_{\Delta^K} p_t(q|x) \sum_a q_a \left\langle x, m_{t,a} \right\rangle dq\right),$$
(33)

using Jensen's inequality. It holds that

$$\int_{\Delta^K} p_t(q|x) \sum_a q_a \left\langle x, m_{t,a} \right\rangle dq \leq \frac{1}{\eta_t} \log \frac{V_{t-1}(x)}{W_t(x)}.$$

Then, we get

$$\sum_{t=1}^T \int_{\Delta^K} p_t(q|x) \left\langle z(q,x), \widehat{\theta}_t \right\rangle dq \leq \sum_{t=1}^T \frac{1}{\eta_t} \left(\log \frac{V_{t-1}(x)}{U_t(x)} + \int_{\Delta^K} p_t(q|x) \psi(-\eta_t \left\langle z(q,x), \widehat{\theta}_t - m_t \right\rangle) dq\right).$$

Noting that $V_0 = U_0$, we have

$$\sum_{t=1}^T \frac{1}{\eta_t} \log \frac{V_{t-1}(x)}{U_t(x)} = \sum_{t=1}^T \frac{1}{\eta_t} \left(\log \frac{V_{t-1}(x)}{V_0} - \log \frac{U_t(x)}{U_0}\right)$$

$$= \sum_{t=1}^{T-1} \left(\frac{1}{\eta_{t+1}} \log \frac{V_t(x)}{V_0} - \frac{1}{\eta_t} \log \frac{U_t(x)}{U_0}\right) - \frac{1}{\eta_T} \log \frac{U_T(x)}{U_0}$$

To bound the first term, we use that $\eta_{t+1} \leq \eta_t$ and an additional application of Jensen's inequality:

$$\frac{1}{\eta_{t+1}} \log \frac{V_t(x)}{V_0} = \frac{1}{\eta_{t+1}} \log \mathbb{E}\left[\exp\left(-\eta_{t+1} \langle \sum_{s=1}^t \widehat{\theta}_s, z(Q_t, x) \rangle\right)\right]$$

$$= \frac{1}{\eta_{t+1}} \log \mathbb{E}\left[\exp\left(-\eta_t \langle \sum_{s=1}^t \widehat{\theta}_s, z(Q_t, x) \rangle\right)^{\frac{\eta_{t+1}}{\eta_t}}\right]$$

$$\leq \frac{1}{\eta_t} \log \mathbb{E}\left[\exp\left(-\eta_t \langle \sum_{s=1}^t \widehat{\theta}_s, z(Q_t, x) \rangle\right)\right] = \frac{1}{\eta_t} \log \frac{U_t(x)}{U_0},$$

Set $Q_{\pi^*(x)} := \{(1 - \frac{1}{T})\pi^*(x) + \frac{1}{T} q | q \in \Delta^K\}$, and denote $q_0$ as the uniform distribution over $K$ arms. We then have

$$U_T(x) \geq \int_{Q_{\pi^*(x)}} \exp\left(-\eta_T \left\langle z(q,x), \sum_{t=1}^T \widehat{\theta}_t \right\rangle\right) dq$$

$$= T^{-K} \int_{\Delta^K} \exp\left(-\eta_T \left\langle z((1-\tfrac{1}{T})\pi^*(x) + \tfrac{1}{T}q, x), \sum_{t=1}^{T} \widehat{\theta}_t \right\rangle\right) dq$$

$$\geq T^{-K} U_0(x) \exp\left(-\eta_T \left\langle z((1-\tfrac{1}{T})\pi^*(x) + \tfrac{1}{T}q_0, x), \sum_{t=1}^{T} \widehat{\theta}_t \right\rangle\right),$$

where the first inequality constitutes a change of variables and the second follows from Jensen's bound. After rearranging and taking the logarithm, we get

$$-\frac{1}{\eta_T} \log \frac{U_T(x)}{U_0(x)} \leq \sum_{t=1}^{T} \left\langle z((1-\tfrac{1}{T})\pi^*(x) + \tfrac{1}{T}q_0, x), \widehat{\theta}_t \right\rangle + \frac{K \log T}{\eta_T}$$

$$= \sum_{t=1}^{T} \left\langle z(\pi^*(x), x), \sum_{t=1}^{T} \widehat{\theta}_t \right\rangle + \frac{1}{T}\sum_{t=1}^{T} \left\langle z(q_0 - \pi^*(x), x), \widehat{\theta}_t \right\rangle + \frac{K \log T}{\eta_T}.$$

Combining everything together, we get

$$\sum_{t=1}^{T} \left(\int_{\Delta^K} p_t(q|x) \left\langle z(q, x), \widehat{\theta}_t \right\rangle dq - \left\langle z(\pi^*(x), x), \sum_{t=1}^{T} \widehat{\theta}_t \right\rangle\right) \leq \sum_{t=1}^{T} \frac{1}{\eta_t} \int_{\Delta^K} p_t(q|x) \psi(-\eta_t \left\langle z(q, x), \widehat{\theta}_t - m_t \right\rangle) dq$$

$$+ \frac{1}{T}\sum_{t=1}^{T} \left\langle z(q_0 - \pi^*(x), x), \widehat{\theta}_t \right\rangle + \frac{K \log T}{\eta_T}.$$

$\square$

From Lemma 4.3 and Lemma 4.4, we get a bound on the second term of (19):

$$\mathbb{E}\left[\sum_{t=1}^{T} \langle Z_t(X_t) - Z^*(X_t), \theta_t \rangle\right] = \mathbb{E}\left[\sum_{t=1}^{T} \left\langle Z_t(X_0) - Z^*(X_0), \widehat{\theta}_t \right\rangle\right]$$

$$\leq \mathbb{E}\left[\sum_{t=1}^{T} \frac{1}{\eta_t} \psi\left(-\eta_t \left\langle Z_t(X_0), \widehat{\theta}_t - m_t \right\rangle\right) + \frac{1}{T}\sum_{t=1}^{T} \left\langle z(q_0 - \pi^*(X_0), X_0), \widehat{\theta}_t \right\rangle + \frac{K \log T}{\eta_T}\right]$$

$$(34)$$

We first find a bound on the first term using Lemma B.4. To satisfy the assumptions of Lemma B.4, we need to show that $\mathbb{E}_t\left[\left(-\eta_t \left\langle Z_t(X_0), \widehat{\theta}_t - m_t \right\rangle\right)^2\right] \leq \frac{1}{100}$:

$$\mathbb{E}_t\left[\left(-\eta_t \left\langle Z_t(X_0), \widehat{\theta}_t - m_t \right\rangle\right)^2\right] = \mathbb{E}_t\left[\eta_t^2 \left(\ell_t(A_t, X_t) - X_t^\top m_{t, A_t}\right)^2 \operatorname{tr}\left(\widetilde{\Sigma}_t^{-1} Z_t(X_0) Z_t(X_0)^\top \widetilde{\Sigma}_t^{-1} Z_t(X_t) Z_t(X_t)^\top\right)\right]$$

$$= \eta_t^2 \mathbb{E}_t\left[\left(\ell_t(A_t, X_t) - X_t^\top m_{t, A_t}\right)^2 \operatorname{tr}\left(\widetilde{\Sigma}_t^{-1} \Sigma_t \widetilde{\Sigma}_t^{-1} Z_t(X_t) Z_t(X_t)^\top\right)\right]$$

$$\leq \eta_t^2 \frac{4}{3} \mathbb{E}_t\left[\left(\ell_t(A_t, X_t) - X_t^\top m_{t, A_t}\right)^2 \operatorname{tr}\left(\widetilde{\Sigma}_t^{-1} \widetilde{\Sigma}_t \widetilde{\Sigma}_t^{-1} Z_t(X_t) Z_t(X_t)^\top\right)\right]$$

$$= \eta_t^2 \frac{4}{3} \mathbb{E}_t\left[\left(\ell_t(A_t, X_t) - X_t^\top m_{t, A_t}\right)^2 Z_t(X_t)^\top \widetilde{\Sigma}_t^{-1} Z_t(X_t)\right]$$

$$\leq \eta_t^2 \mathbb{E}_t\left[\left(\ell_t(A_t, X_t) - X_t^\top m_{t, A_t}\right)^2 Z_t(X_t)^\top \Sigma_t^{-1} Z_t(X_t)\right]$$

$$\leq dK \eta_t^2 \gamma^2 \mathbb{E}_t\left[\left(\ell_t(A_t, X_t) - X_t^\top m_{t, A_t}\right)^2\right] \qquad (35)$$

$$\leq \frac{1}{100}, \qquad (36)$$

where the first inequality follows from $\ell_t \leq 1$ and (29), the second is immediate from (29) and the fact that for symmetric positive definite matrices $A \succeq B$ follows from $B^{-1} \succeq A^{-1}$. The third

inequality follows from the truncation in the algorithm and the last is immediate from plugging in the definition of $\eta_t$. So, by applying Lemma B.4 and (35), we get:

$$\frac{1}{\eta}\mathbb{E}\left[\psi\left(-\eta\left\langle Z_t(X_0),\widehat{\theta}_t - m_t\right\rangle\right)\right] \leq \frac{2}{\eta}\mathbb{E}\left[\left(-\eta\left\langle Z_t(X_0),\widehat{\theta}_t - m_t\right\rangle\right)^2\right] \leq 2dK\eta\gamma^2\mathbb{E}_t\left[\left(\ell_t(A_t,X_t) - X_t^\intercal m_{t,A_t}\right)^2\right].$$
(37)

For the second term of (34), we simply get from $-1 \leq \ell_t \leq 1$ and $\widehat{\theta}_t$ is unbiased:

$$\mathbb{E}\left[\frac{1}{T}\sum_{t=1}^T\left\langle z(q_0 - \pi^*(X_0),X_0),\widehat{\theta}_t\right\rangle\right] = \mathbb{E}\left[\frac{1}{T}\sum_{t=1}^T\left\langle z(q_0 - \pi^*(X_0),X_0),\theta_t\right\rangle\right] \leq 2. \quad (38)$$

The expression that we use for the learning rate is the following:

$$\eta_t = (100dK\gamma^2 + d(\widehat{V}_{t-1} + 1 + G_t)))^{-1/2},$$

where $G_t = 8\sqrt{2\widehat{V}_{t-1}\ln T + 144\ln^2 T} + 176\ln T$. We show that with probability at least $1 - \delta$ the following holds for all $t \in [T]$:

$$V_t \leq \widehat{V}_t + 8\sqrt{\widehat{V}_t\ln(2T/\delta) + 72\ln(2T/\delta)^2} + 88\ln(T/\delta) \quad (39)$$

Let $Y_s = \mathbb{E}_s\left[\left(\ell_t(A_t,X_t) - X_t^\intercal m_{t,A_t}\right)^2\right] - \left(\ell_t(A_t,X_t) - X_t^\intercal m_{t,A_t}\right)^2$. Then, $Y_s \leq 4$ almost surely, since $\ell_t(A_t,X_t) - X_t^\intercal m_{t,A_t} \leq 2$. Similarly we bound the second moment of $Y_s$, using Jensen's inequality:

$$\mathbb{E}_s\left[Y_s^2\right] = \mathbb{E}_s\left[\left(\mathbb{E}_s\left[\left(\ell_t(A_t,X_t) - X_t^\intercal m_{t,A_t}\right)^2\right] - \left(\ell_t(A_t,X_t) - X_t^\intercal m_{t,A_t}\right)^2\right)^2\right]$$

$$\leq 2\mathbb{E}_s\left[\left(\ell_t(A_t,X_t) - X_t^\intercal m_{t,A_t}\right)^2\right]^2 + 2\mathbb{E}_s\left[\left(\ell_t(A_t,X_t) - X_t^\intercal m_{t,A_t}\right)^4\right]$$

$$\leq 16\mathbb{E}_s\left[\left(\ell_t(A_t,X_t) - X_t^\intercal m_{t,A_t}\right)^2\right].$$

By Lemma B.2, the following holds for some $\delta' \in (0,1)$:

$$V_t \leq \widehat{V}_t + 8\max\left\{2\sqrt{V_t},\sqrt{\ln(1/\delta')}\right\}\sqrt{\ln(1/\delta')} \quad (40)$$

Note that this inequality i can be rearranged as

$$V_t \leq \widehat{V}_t + 8\sqrt{\widehat{V}_t\ln(1/\delta') + 72\ln(1/\delta')^2} + 88\ln(1/\delta').$$

Then, taking a union bound over $t \in [T]$ and taking $\delta = \delta'/T$, we get that (39) holds for all $t \in [T]$. Let $\mathcal{E}_T$ be an event that for all $t \in [1,T]$, (39) holds with $\delta = 1/T$. From (37), (38), and the choice of $\eta_t$, we get:

$$R_T \leq \mathbb{E}\left[2dK\gamma^2\sum_{t=1}^T\eta_t\left(\ell_t(A_t,X_t) - X_t^\intercal m_{t,A_t}\right)^2 + 2 + \frac{K\log T}{\eta_T}\right] \quad (41)$$

$$= 2dK\gamma^2\mathbb{E}\left[\sum_{t=1}^T\eta_t\left(\ell_t(A_t,X_t) - X_t^\intercal m_{t,A_t}\right)^2\mathbb{1}\left\{\mathcal{E}_T\right\}\right]$$

$$+ 2dK\gamma^2\mathbb{E}\left[\sum_{t=1}^T\eta_t\left(\ell_t(A_t,X_t) - X_t^\intercal m_{t,A_t}\right)^2\mathbb{1}\left\{\overline{\mathcal{E}}_T\right\}\right] + 2 + \mathbb{E}\left[\frac{K\log T}{\eta_T}\right]$$

$$\leq 2\sqrt{d}K\gamma^2\sum_{t=1}^T\frac{V_t - V_{t-1}}{\sqrt{V_t}} + \frac{1}{T}2\sqrt{d}K\gamma^2 T + 2 + \frac{K\log T}{\eta_T'}$$

$$= 2\sqrt{d}K\gamma^2\sum_{t=1}^T\frac{(\sqrt{V_t} - \sqrt{V_{t-1}})(\sqrt{V_t} + \sqrt{V_{t-1}})}{\sqrt{V_t}} + 2\sqrt{d}K\gamma^2 + 2 + \frac{K\log T}{\eta_T'}$$

$$\leq 4\sqrt{d}K\gamma^2 \sum_{t=1}^{T}(\sqrt{V_t} - \sqrt{V_{t-1}}) + 2\sqrt{d}K\gamma^2 + 2 + \frac{K\log T}{\eta_T'}$$

$$\leq 4\sqrt{d}K\gamma^2\sqrt{V_T} + 2\sqrt{d}K\gamma^2 + 2 + \frac{K\log T}{\eta_T'}.$$

which implies the result of Theorem 3.1. In the equation above, $\eta_T' = (100dK\gamma^2 + d(V_{t-1} + 1 + G_t')))^{-1/2}$ and $G_t' = 8\sqrt{2V_{t-1}\ln T + 144\ln^2 T + 176\ln T}$. In line 4 we used that $\mathbb{E}\left[1/\eta_T\right] \leq \mathbb{E}\left[1/\eta_T'\right]$ by Jensen's inequality to show that

$$\mathbb{E}\left[\frac{1}{\eta_T}\right] = \mathbb{E}\left[(100dK\gamma^2 + dK(\widehat{V}_{t-1} + 1 + G_t))^{1/2}\right] \leq (100dK\gamma^2 + d(V_{t-1}+1+G_t'))^{1/2} = \frac{1}{\eta_T'}.$$

$\square$

**Proof of Theorem 3.2**  As it was done in the proof of Theorem 3.1 we control the deviation of the learning rate

$$\eta_t = (100dK\gamma^2 + d(\widehat{L}_{t-1} + 1 + H_t)))^{-1/2},$$

where $H_t$ is as defined in (14). Using Lemma B.2, we show that with probability at least $1 - \delta$ the following holds for all $t \in [T]$:

$$L_t \leq \widehat{L}_t + 8\sqrt{\widehat{L}_t \ln(1/\delta)} + 20\ln(2T/\delta)^2 + 36\ln(2T/\delta) \tag{42}$$

$D_s = \mathbb{E}_s\left[\langle X_s, \theta_{s,A_s}\rangle\right] - \langle X_s, \theta_{s,A_s}\rangle$. Then, $D_s \leq 2$ almost surely and by Jensen's inequality

$$\mathbb{E}_s\left[D_s^2\right] = \mathbb{E}_s\left[(\mathbb{E}_s\left[\langle X_s, \theta_{s,A_s}\rangle\right] - \langle X_s, \theta_{s,A_s}\rangle)^2\right] \leq 2\mathbb{E}_s\left[\langle X_s, \theta_{s,A_s}\rangle\right]^2 + 2\mathbb{E}_s\left[(\langle X_s, \theta_{s,A_s}\rangle)^2\right] \leq 4\mathbb{E}_t\left[\ell_t(X_t, A_t)\right].$$

By Lemma B.2, the following holds for some $\delta' \in (0, 1)$:

$$L_t \leq \widehat{L}_t + 4\max\left\{2\sqrt{L_t}, \sqrt{\ln(1/\delta')}\right\}\sqrt{\ln(1/\delta')} \tag{43}$$

which can be rearranged as

$$L_t \leq \widehat{L}_t + 8\sqrt{\widehat{L}_t \ln(1/\delta')} + 20\ln(1/\delta')^2 + 36\ln(1/\delta').$$

Then, taking a union bound over $t \in [T]$ and taking $\delta = \delta'/T$, we get that (42) holds for all $t \in [T]$. Let $\mathcal{E}_T$ be an event that for all $t \in [1, T]$, (42) holds with $\delta = 1/T$. From (37), (38), the choice of $\eta_t$, $m_t = \bar{0}$ and since $0 \leq \ell_t \leq 1$, we get:

$$R_T \leq \mathbb{E}\left[2dK\gamma^2 \sum_{t=1}^{T} \eta_t \ell_t^2(A_t, X_t) + 2 + \frac{K\log T}{\eta_T}\right] \leq \mathbb{E}\left[2dK\gamma^2 \sum_{t=1}^{T} \eta_t \ell_t(A_t, X_t) + 2 + \frac{K\log T}{\eta_T}\right]$$

$$= 2dK\gamma^2 \mathbb{E}\left[\sum_{t=1}^{T} \eta_t \ell_t(A_t, X_t)\mathbb{1}\left\{\mathcal{E}_T\right\}\right] + 2dK\gamma^2\mathbb{E}\left[\sum_{t=1}^{T} \eta_t \ell_t^2(A_t, X_t)\mathbb{1}\left\{\overline{\mathcal{E}}_T\right\}\right] + 2 + \mathbb{E}\left[\frac{K\log T}{\eta_T}\right]$$

$$\leq 2\sqrt{d}K\gamma^2 \sum_{t=1}^{T} \frac{L_t - L_{t-1}}{\sqrt{L_t}} + \frac{1}{T}2\sqrt{d}K\gamma^2 T + 2 + \frac{K\log T}{\eta_T'}$$

$$= 2\sqrt{d}K\gamma^2 \sum_{t=1}^{T} \frac{(\sqrt{L_t} - \sqrt{L_{t-1}})(\sqrt{L_t} + \sqrt{L_{t-1}})}{\sqrt{L_t}} + 2\sqrt{d}K\gamma^2 + 2 + \frac{K\log T}{\eta_T'}$$

$$\leq 4\sqrt{d}K\gamma^2 \sum_{t=1}^{T}(\sqrt{L_t} - \sqrt{L_{t-1}}) + 2\sqrt{d}K\gamma^2 + 2 + \frac{K\log T}{\eta_T'}$$

$$\leq 4\sqrt{d}K\gamma^2\sqrt{L_T} + 2\sqrt{d}K\gamma^2 + 2 + \frac{K\log T}{\eta_T'},$$

where in the equation above, $\eta_t' = (100dK\gamma^2 + d(L_{t-1} + 1 + H_{t-1}')))^{-1/2}$ and $H_t' = 8\sqrt{2L_{t-1}\ln T + 40\ln T} + 72\ln T$. By solving the quadratic equation over $L_T^*$, we obtain the statement of the theorem.

$\square$

# D   On the difference between LINEXP3 and CONTEXTEW

Consider the LINEXP3 algorithm of Neu and Olkhovskaya (2020), that draws actions after observing the context $X_t$ with probability

$$\pi_t\left(a|X_t\right) = (1-\gamma)\frac{w_t(X_t,a)}{\sum_{a'} w_t(X_t,a')} + \frac{\gamma}{K},$$

where $w_t(X_t,a) = \exp\left(-\eta \sum_{s=0}^{t-1}\langle X_t, \widehat{\theta}_{s,a}\rangle\right)$ and using the estimator

$$\widetilde{\theta}_{t,a}^* = \mathbb{1}\left\{A_t = a\right\} S_{t,a}^{-1} X_t \left\langle X_t, \theta_{t,a}\right\rangle,$$

where $S_{t,a} = \mathbb{E}_t\left[\pi_t(a|X_t)X_t X_t^\intercal\right]$. Since LINEXP3 uses implicit exploration with probability $\gamma$, $\lambda_{min}(S_{t,a}) \geq \lambda_{min}(\Sigma)\frac{\gamma}{K}$. But then, setting $\gamma = 0$, $S_{t,a}$ is still invertible as no actions have $\pi_t\left(a|X_t\right) = 0$. But still, the smallest eigenvalue $\lambda_{\min}(S_{t,a})$ can be arbitrary small. Then, the analysis of the variance term in LINEXP3 looks as:

$$\mathbb{E}_t\left[\sum_{a=1}^{K}\pi_t(a|X_0)\langle X_0,\widehat{\theta}_{t,a}\rangle^2\right] = \mathbb{E}_t\left[\sum_{a=1}^{K}\pi_t(a|X_0)\left(X_0^\intercal S_{t,a}^{-1}X_t X_t^\intercal \theta_{t,a}\mathbb{1}\left\{A_t=a\right\}\right)^2\right]$$

$$= \mathbb{E}\left[\ell_t(X_t,A_t)^2\text{tr}\left(\pi_t(a|X_0)X_0 X_0^\intercal S_{t,a}^{-1}X_t X_t^\intercal S_{t,a}^{-1}\right)\right].$$

We can define $Var_t'$ for LINEXP3 in direct analogy to $Var_t$ for CONTEXTEW above, which gives (almost surely):

$$\mathbb{E}_{X_0}\left[Var_t'\right] = \mathbb{E}_{X_0}\left[\text{tr}\left(\pi_t(a|X_0)X_0 X_0^\intercal S_{t,a}^{-1}X_t X_t^\intercal S_{t,a}^{-1}\right)\right]$$

$$= \mathbb{E}_{X_0}\left[\text{tr}\left(\Sigma_{t,a}S_{t,a}^{-1}X_t X_t^\intercal S_{t,a}^{-1}\right)\right] = X_t^\intercal S_{t,a}^{-1}X_t,$$

which can be arbitrary large.

Meanwhile, the smallest eigenvalue $\lambda_{\min}(\Sigma_{t,a})$ can be arbitrary small too. But, as we showed above in the analysis of CONTEXTEW, $Var_t$ is bounded by $dK\gamma^2$ because of the log-concavity of $Z_t(X_t)$ and step (6) of CONTEXTEW.

