# OpenReview forum: "First- and Second-Order Bounds for Adversarial Linear Contextual Bandits"
_NeurIPS.cc/2023/Conference — NeurIPS 2023 poster_

### Official Review · Reviewer_pxmm · 2023-06-19

**Soundness:** 3 good
**Presentation:** 3 good
**Contribution:** 3 good
**Rating:** 6
**Confidence:** 4

**Summary:**

This paper studies the adversarial linear contextual bandit setting, an online learning problem with $K$ arms, stochastic contexts in $\mathbb{R}^d$, and adversarial rewards (which are assumed to be a linear function of the context for each arm). Previous work proves a worst-case regret guarantee that is $\sqrt{KdT}$. They obtain regret guarantees which instead of scaling with $\sqrt{T}$ can scale with other quantities that are smaller: a certain "second-order" quantity called $V_T$, and a first-order or (small loss) bound of $L_T^\star$.

This is accomplished using a new algorithm called ContextEW which is a continuous exponential weight style algorithm, inspired by previous work of [Ito et al. '20]. They show that it achieves the aforementioned regret guarantees and can be efficiently implemented.

**Strengths:**

- The paper is generally well written and explains the problem statement and results well.
- The paper provides a new algorithm for adversarial linear contextual bandits which achieves first and second order bounds, a standard object of study in bandit literature.

**Weaknesses:**

- It is a bit unclear from me how much of the work is original, or whether it is a straightforward exercise to modify the results from [Ito et al. '20] and [Neu-Olkhovskaya '20]. Perhaps this can be made clearer in the proof sketch and technical sections to delineate what techniques/results are new.
- The discussion on computational complexity could be improved (see questions below).
- The key intuitions for the algorithm and analysis are unclear to me (see questions below).
- The results rely on the assumption that the distribution of the contexts is log-concave (which is a limitation that comes from building upon previous work).

**Questions:**

1. In the equation after line 131, what exactly is $\mathbb{E} \theta_{t,a}$? The $\theta$ are assumed to be adversarially selected, under some choices of actions selected by a learner, so what does it mean to take the expectation here?
2. What is the intuition for the truncation/rejection sampling step? It is mentioned that it is used to reduce the variance. What would the final bound look like if there was no truncation/rejection sampling step?
3. Typo: in (10), $P_t$ should be $\mathbb{P}_t$ to be consistent with the rest of the notation.
4. Computational complexity:
    - I don't really understand how one should calculate $\tilde{\Sigma}$. This requires a sampling oracle to $\mathcal{D}$ in order to use the [Lovasz-Vempala '07] results right?  My understanding is that one can try to approximate (9) using draws from $\mathcal{D}$. Can the authors explain how to (approximately) calculate  $\tilde{\Sigma}_{t,a}$?
    - A follow up to the previous question is how does the error in sampling/computing the covariances propagate to the final error bound?
    - One can try to pick $\epsilon = 1/\mathrm{poly}(T)$. Is computing the covariance still "computationally efficient" since the computational complexity seems to scale polynomially with the total length of the problem $T$?
5. At a high level, what is the motivation for studying second-order bounds? When would we expect these to be small in practice?
6. Questions about proof details:
    - typo: I believe in Lemma C.1 there should not be a subscript of $Q_t$ in line 447.
    - In the last inequality after line 468, where did the integration over $x$ go?
    - I'm a bit confused about why the quantity $m_t$ appears in the Proof of Lemma C.4. For example, in the first equality after 495, how does the $-m_t$ appear? I'm not able to follow this. Is there some relationship between $\hat{\theta}_t$ and $m_t$ that is being exploited in this proof?
    - I didn't follow (38), how is the unbiasedness of $\hat{\theta}_t$ used?
7. It would be nice if the relevant lemmas from [Ito et al. '20] were restated in the appendix, to make the proof self contained. For example, Lemma 6 of their paper is used in the proof.


**Limitations:**

None.

---

> ### Author Rebuttal · Authors · 2023-08-10
>
> To emphasize the above, an application of an algorithm designed to solve the linear bandit problem cannot be applied directly to the linear contextual bandit, and there are no such examples for this setting.
> With regard to your questions:
>
> 1.	The meaning of the expectation is with respect to the filtration (i.e. the adversary can pick the parameter $\theta$ based on previous random events), or any randomness the adversary may add. Note that our ability to derive a first-order bound is an implicit assumption that the adversary may not be worst-case.
>
> 2.	As we state in the discussion section, without the rejection sampling step,  it can be removed by paying a $\log(1/\lambda_{\min}(\Sigma))$ multiplicative
> term in the regret by implementing additional exploration with probability $1/T$.
>
> 3.	Thanks for spotting the typo, we will fix it.
>
> 4. You are correct - it is required to have an oracle for taking samples of $\mathcal{D}$, but we have assumed that the distribution is known to the learner, and that the learner has oracle access. $\tilde\Sigma_{t,a}$ can be calculated by drawing samples from the joint distribution of $\mathcal{D}$ and $\tilde{\pi}_{t}(\cdot|X)$ $m \geq 100 \frac{dK}{\epsilon^{2}}\mathrm{log}^{3}\frac{dK}{\epsilon^{2}}$ times (Theorem 2.6 and Corrolary 2.7 from L. Lovász and S. Vempala. (2007)). This is where log-concavity of both $\mathcal{D}$ and $\tilde{\pi}$ helps us considerably, as their proof heavily relies on this assumption. Indeed this gives us a computational complexity of $O(dKT^{2})$ per round; a vast improvement over the MYGA algorithm which achieves the same regret bound (and the best achieved by any algorithm for this problem setting; our Theorem 2.1), which has a complexity of $O(T^{dK})$ per round, and bringing our algorithm well within the reaches of implementation.
>
> 5.	The variance-dependent bound is advantageous when a good prediction of the losses are available, and thus
> the actual regret incurred by the algorithm is small. Another example is when the losses are stochastic and have small
> variance.
>
> 6.
> R: Typo: I believe in Lemma C.1 there should not be a subscript of $Q_t$ in line 447:
>
> This is right, we will fix this typo.
>
> R: In the last inequality after line 468, where did the integration over $x$ go?
>
>  There is a typo - sign of integration over X and dx are missing.
>
> R: I'm a bit confused about why the quantity $m_t$
> 	appears in the Proof of Lemma C.4. For example, in the first equality after 495, how does the $-m_t$
> 	appear? I'm not able to follow this. Is there some relationship between $\hat{\theta}_t$
> 	and $m_t$
> 	that is being exploited in this proof?
>
>  Indeed, there is  a typo in our definition of $w_t$. The correct definition is $w_t(q|x) = \exp(- \eta_t \sum_{a=1}^K q_a \langle X_t, \sum_{s=1}^{t-1}\hat{\theta}_{s,a} + m_{t,a}\rangle)$.
>
> R: I didn't follow (38), how is the unbiasedness of $\hat{\theta}_t$ used?
>
> We use $\mathbb{E}[\langle z(q_0 - \pi^*(X_0), X_0), \hat{\theta}_t\rangle ] = \mathbb{E}[\langle z(q_0 - \pi^*(X_0), X_0), \theta_t\rangle ]  \le 2$, where in the last step we used that the loss values are $\ell_t \in [-1,1]$. We will add this explanation to the analysis.
>
> 7. We will add the corresponding lemmas to the analysis.

---

> > ### Comment · Reviewer_pxmm · 2023-08-14
> > **Thanks**
> >
> > Thank you for your detailed responses to my questions. I have no further questions at this point.

---

> > > ### Author Response · Authors · 2023-08-21
> > >
> > > Thank you for the detailed review!

---

### Official Review · Reviewer_zjgE · 2023-06-27

**Soundness:** 3 good
**Presentation:** 3 good
**Contribution:** 1 poor
**Rating:** 5
**Confidence:** 4

**Summary:**

The submission considers semi-adversarial contextual bandit setting --- where d-dimensional contexts x_t are drawn iid, where some opponent picks K loss functions that are linear in the contexts at the beginning of each round, before the actual context x_t is drawn, and where he learner picks one of the K loss functions and suffers the corresponding linear loss. This setting was introduced by Neu and Olkhovskaya (COLT 2020), who provided 'zero-order' regret bounds of order \sqrt{K d T}. The present submission refines the regret bounds into first-order (for non-ngative losses, as usual) and second-order regret bounds, involving respectively the cumulative loss of the best static policy and the expected sum of instantaneous variances of the rewards obtained by the algorithms (and generalizations thereof). This is achieved by considering a continuous version of exponential weights, à la Cover (1991), and by leveraging techniques introduced by Ito, Hirahara, Soma, and Yoshida (Neurips 2020) for such refined first-order and second-order bounds in the setting of non-contextual adversarial linear bandits.

The key to the results is the reduction (18) of this semi-adversarial contextual bandit problem into a linear bandit problem, as underlined in lines 285-288.


**Strengths:**

The submission has a clear technical focus, and combines in a neat way techniques from Neu and Olkhovskaya (COLT 2020) and Ito, Hirahara, Soma, and Yoshida (Neurips 2020). The proof sketch provided on pages 6--8 is well structured. Up to some typos and some unclear pieces of notation, I can only praise the writing.


**Weaknesses:**

My main issue is with the algorithm, which relies on distributions that are difficult to compute exactly---as always with continuous exponential weights (see articles from Vempala and co-authors at the end of the 90s, commenting on Cover's original contributions). Allowing \epsilon--approximations reduces the computational burden (see Section 3.3) but the effect of using these \epsilon--approximations on the regret bounds is not discussed. Would this add some \epsilon T terms? Then, it would probably be wise to pick \epsilon = 1/T, not to kill with this extra term the possible fast rates of convergence yielded by first- and second-order bounds. (Also, it is not clear to me from Section 3.3 whether these \epsilon--errors propagate and get worse and worse as t increases, or if for each t, we may and should compute an \epsilon--approximation from scratch, not based on previous probability distributions.)

Another issue would be that I am unsure of what the second-order bound of Theorem 3.2 brings: it is highly general, and no corollary or example for the functions m_t are provided (except, I guess, the choices m_t \equiv 0). More generally, I am unsure of how important it is to get refined bounds with a rather non-practical algorithm based on continuous exponential weights.

A more minor issue is about the assumption of log-concavity. It is satisfied by many distributions (see https://en.wikipedia.org/wiki/Logarithmically_concave_function#Log-concave_distributions, and the submission could give examples) but is also not satisfied by other important ones.

Typos / Remarks
- Lines 96-97: rather cite Cover (1991) or to the very least the monograpu by Cesa-Bianchi and Lugosi (2006), instead of a 2018 reference
- Line 123: I believe you refer to it as \pi(X_t) and never use \pi(a|X_t)
- Line 131: Yes, this seems clear indeed
- Line 134: I would rather have written E_{t-1} instead of E_t
- Line 144 / Eq. (4); see also later occurrences like in Th.3.2: Is L^*_T the same as L_T^{\pi^*}?
- Algorithm 1: Say that each m_s is a function that is measurable w.r.t. \mathcal{F}_{s-1}
- Eq.(6): Somewhere earlier than in lines 291 and following, discuss the number of steps needed to achieve (6), i.e., geometric distribution with a parameter that you anyway bound in the proof---but mention it early, at latest on page 5 or in Section 3.3
-  Line 185: Define somewhare what log-concavity of a distribution is, and provide examples
- Eq. (18) and comments in lines 285-288: This is the key step and it's somewhat hidden, I'd suggest to put it more in front
- Lemma 4.2: Notation is heavy and the result is extremely difficult to read, is there any way to make it smoother? That part and Eq. (19) should also be put more in front.
- Line 291: Missing word before 1 ('Algorithm' I guess)
- Appendices: Cauchy-Schwarz without a 't'

**Questions:**

Please comment on the main two weaknesses that I listed.

**Limitations:**

The authors are clear about the limitations, and in particular, on dependency of their regret bounds on K, which does not match, in the worst-case, the known \sqrt{d K T} regret bound. The discussion on page 9 is complete and upfront about the scope of their contribution.

---

> ### Author Rebuttal · Authors · 2023-08-10
>
>
> Indeed, we did omit a discussion of the epsilon approximations in the regret bound. As you note, one would require $\epsilon = O(1/T)$ to ensure that the contribution of the additive approximation sums over rounds to a constant. It is also correct that the epsilon approximation should be recalculated per round as the covariance matrix depends on the current policy. The assumption of log-concavity is a restrictive one, and we are making efforts to remove it in further work on this topic.
>
>
> With regard the choice of $m_{t}$ for $t\in[T]$, any sequence of $m_{t}$ will satisfy the bounds we've given (note that $m_{t}$ is in the definition of $V_{T}$), but some choices are better than others. The obvious choice being $m_{t}=0$ for all $t\in[T]$, which leads to a first-order bound. In case we have some prior information about the losses, we have room to encode this in $m_{t}$, and if not, we can use $\mathcal{F}_{t-1}$-measurable mean estimates, or ridge regression as described by Ito to obtain a second-order result. Since the ridge regression regret is $O(\mathrm{log}(T))$, this does not affect the rate significantly.
>
> Many thanks for your careful reading, and consequent identification of typos/points of improvement for the camera-ready version.

---

> > ### Comment · Reviewer_zjgE · 2023-08-11
> >
> > I acknowledge reading the entire thread of reviews and corresponding rebuttals. I confirm my initial evaluation.

---

> > > ### Author Response · Authors · 2023-08-21
> > >
> > > Thank you for the feedback!

---

### Official Review · Reviewer_QEEf · 2023-06-30

**Soundness:** 3 good
**Presentation:** 3 good
**Contribution:** 2 fair
**Rating:** 5
**Confidence:** 4

**Summary:**

The authors explore the adversarial linear contextual bandit problem, where  the context is i.i.d. drawn from a known fixed log-concave distribution, and the linear loss functions are adaptive adversaries.
They provide first adaptive bounds for this setting, including a first-order bound based on the cumulative loss of the optimal policy and a second-order bound based on the squared losses of the algorithm. These bounds demonstrate improvements over the worst-case bound in scenarios where the environment is  'nice'.

**Strengths:**

The authors first provide the adaptive bound for adversarial linear contextual bandits, which improves the worst case bound in benign environments.
The computational complexity of the algorithm is commendable, particularly in moderate environments.
The results and the addressed problem hold substantial significance and are likely to capture the interest of a broad audience in Neurips.
Furthermore, the authors thoroughly discuss future directions, which have the potential to inspire further research aimed at improved adaptive bounds in linear bandits setting.

**Weaknesses:**

I have two primary concerns regarding this work and would like to request the authors to provide additional clarifications.

==Weakness 1==

It appears that the results and techniques heavily rely on previous work, such as

Ito, S., Hirahara, S., Soma, T., and Yoshida, Y. (2020). Tight first- and second-order regret bounds for adversarial linear bandits,

which first introduce adaptive bounds for adversarial linear bandits.
In comparison to the adversarial linear bandits setting, the authors in this paper consider the case with i.i.d. contexts and introduce some additional assumptions, such as known context distribution and log-concavity. These assumptions may simplify the problem and restrict the practical application of the proposed algorithms. I suggest the authors emphasize the technical contribution of their paper, as this aspect is not clearly highlighted in the "Techniques" subsection in the Introduction.

==Weakness 2==

There does not seem to be any experiments of any type.
Neither the first-order bound nor the second-order bound demonstrate strict improvements over the worst-case bound.
In other words, they may not be tight enough.
In this case, I kindly recommend that the authors provide numerical results, which would enable readers to gain a better understanding of the algorithm's behavior.  While numerical evaluations may not significantly enhance the primary theoretical message of the paper, they would improve the overall completeness of the work, considering that linear bandits are closely related to practical applications.

**Questions:**

Please see the points mentioned in the weaknesses above and provide comments on them.

**Limitations:**

I don't see any issue on societal impact.

---

> ### Author Rebuttal · Authors · 2023-08-10
>
> We want to emphasize that algorithms designed
> for solving the linear bandit problem cannot be applied to the
> contextual bandit problem, and thus there is no previous method that
> could simply yield the result that we claim in this paper. For a comparison with Ito et al., please see our general reply to all
> reviewers.
>
> While this work is primarily theoretical, we understand the allure of supporting experiments. However, given our main contribution is theoretical in nature, this paper remains comprehensive even in the absence of experiments well within the standards of this line of work.

---

> > ### Comment · Reviewer_QEEf · 2023-08-17
> > **Response to Rebuttal**
> >
> > Thank you for your response and comments. While I acknowledge the paper's interesting contributions, I still consider it borderline for acceptance. Therefore, I would prefer to maintain my current score.

---

> > > ### Author Response · Authors · 2023-08-21
> > >
> > > Thank you for your review!

---

### Official Review · Reviewer_hPEo · 2023-07-06

**Soundness:** 3 good
**Presentation:** 3 good
**Contribution:** 2 fair
**Rating:** 6
**Confidence:** 4

**Summary:**

Addressing to the K-armed linear contextual adaptive adversary bandit problem with d-dimensional contexts over T rounds, using a truncated version of the continuous exponential weights algorithm over the probability simplex, the expected regret about the cumulative second moments of the losses is obtained. and then the first order bound is derived.

**Strengths:**

generalizing the results of tight First- and Second-Order Regret Bounds for Adversarial Linear Bandits to First- and Second-Order Bounds for Adversarial Linear Contextual Bandits.

**Weaknesses:**

1 The results of the paper heavily rely on a combination of the recently proposed techniques of Ito et al. (2020) for linear bandits with tools designed by Neu and Olkhovskaya (2020) to deal with the contextual case.
2 The conclusion only applies to the scenario of the joint log-concavity of the context distributions and the exponential weights posterior over the simplex of actions, results may not be generalizable.

**Questions:**

1 if the distribution for contexts is time-vary or need to be estimated from data, what will the situation be like?
2 the analysis is all based on the assumption that they can access the distribution for contexts, which is not the case for practical case.  How to handle this difficulty, please give further discussion.

**Limitations:**

the authors adequately addressed the limitations

---

> ### Author Rebuttal · Authors · 2023-08-10
>
> Please see our general reply for
> answers to some of the points you raised. Here we only respond to the
> remaining points.
>
> Regarding time-varying distributions: it is common in contextual bandits
> to assume that both the contexts and the losses for the online recommendation arms are i.i.d.
> It is an important strength of our setting to relax this to allow for
> adversarial (linear) losses for the arms. On the other hand, it is known to be impossible to fully remove the i.i.d.~
> assumption on the contexts: if the contexts are adversarial, then sublinear regret cannot be achieved, since one can map
> the problem to that of classification with threshold functions at zero margin (see Corollary 6 from Ben-David,
> S., Pál, D., and Shalev-Shwartz, S. (2009). Agnostic online learning. with $\mu =
> O(\sqrt{T})$, and the matching lower bounds). The intermediate regime of
> e.g.\ bounded step-wise divergence between successive context
> distributions leaves an interesting question worthy of further study,
> but is likely nontrivial.

---

> > ### Author Response · Authors · 2023-08-10
> > **Correction**
> >
> > Should be $1/\mu =O(\sqrt{T})$.

---

### Author Rebuttal · Authors · 2023-08-09


To all reviewers:

We thank all reviewers for their insightful comments.

Comparison to Neu \& Olkhovskaya (2020): We note that our approach is
not a straight-forward extension of Neu \& Olkhovskaya (2020), which
would lead to discrete exponential weights over arms (i.e.\ the corners
of the simplex) rather than continuous exponential weights over the full
simplex. In fact, it was a breakthrough in our investigations to make
this switch to the continuous version, because all variants of the
discrete approach that we tried ended up having too large variance to
obtain first- or second-order bounds.


Comparison to Ito et al. 2020:  It's crucial to note that algorithms for the linear bandit problem are not
universally applicable to the contextual bandit problem, rendering the method used in this paper unique. Our idea of drawing
the distribution over arms from a context-adapted version of continuous exponential weights, and only then sampling an
action from the resulting simplex element is clearly not a trivial modification of the linear bandit algorithm, and has
not been done before in the existing literature. To restate, we are proposing a conditional reduction of linear
contextual bandits to the linear bandit scenario when contexts are i.i.d. We establish the effectiveness of this
reduction by demonstrating that the former can take on the data-dependent regret bound of the latter. This may pave the way for an area of inquiry within the bandit literature, as
this could also be extended to broader function approximation scenarios where adaptive regret bounds are crucial. Consequently, this approach could have potential applications across several subfields of decision-theoretic
learning.

Assumptions: While our log-concave assumption on the distribution may not be ideal, we find it remarkable that such a simple assumption can result in such a big improvement in the regret guarantees. While we agree that this assumption is arguably strong, we also wish to point out that it encompasses many natural distributions such as Gaussian or uniform on a convex set. Making such (and even more specific) assumptions about otherwise unknown distributions is not uncommon in learning theory (see, e.g., the numerous papers that have been written about stochastic multi-armed bandits with Gaussian rewards), so we feel that our result may very well be interesting enough for the bandit community, especially given the attention that the problem has been receiving recently. Indeed, we wish to reiterate our main point that obtaining first-order regret bounds for contextual bandits is a highly nontrivial problem that has been considered by many research groups over the past years, and thus any progress made on the topic is of potentially significant interest. We hope that our contributions may inspire future progress on this problem, and that our assumptions can eventually be relaxed.

Regarding the known context distribution: this is an assumption that is
present already in the work of Neu \& Olkhovskaya, and this is still an open problem to get rid of this assumption in
order to obtain $\tilde O(\sqrt{dKT})$ regret bound.  It can be justified
in cases where labels/losses are expensive to obtain, but obtaining unlabeled contexts is cheap (as can be the case,
e.g., in online advertising).  As discussed in Section 3.3, it is sufficient to have sampling access to
the context distribution.

---

### Decision · Program_Chairs · 2023-09-21

**Decision:**

Accept (poster)

**Comment:**

The authors make a substantial theoretical contribution to the area of contextual bandits, proving stronger regret bounds for the case of adversarial linear contextual bandits, that depend on adaptive properties of the realized sequence of losses.

The reviewers all agree that this is an interesting and new theoretical results. There is only one concern regarding incrementality to prior work in terms of techniques. The authors offered several dimensions in which their techniques differ in their rebuttal responses. It would be great if these clarifications could be made more vivid in their paper if the paper gets accepted.